# High contribution of anthropogenic combustion sources to atmospheric inorganic reactive nitrogen in South China evidenced by isotopes

Tingting Li[1,2,4], Jun Li[*1,2], Zeyu Sun[3,4], Hongxing Jiang[1], Chongguo Tian[3], Gan Zhang[1,2]

[1]State Key Laboratory of Organic Geochemistry and Guangdong province Key Laboratory of Environmental Protection and Resources Utilization, Guangdong-Hong Kong-Macao Joint Laboratory for Environmental Pollution and Control, Guangzhou Institute of Geochemistry, Chinese Academy of Sciences, Guangzhou, 510640, China

[2]CAS Center for Excellence in Deep Earth Science, Guangzhou 510640, P. R. China

[3]Yantai Institute of Coastal Zone Research, Chinese Academy of Sciences, Yantai 264003, P. R. China

[4]University of Chinese Academy of Sciences, Beijing 100049, P. R. China

*Correspondence to*: Jun Li (junli@gig.ac.cn)

**Abstract:** Due to the intense release of reactive nitrogen (Nr) from anthropogenic activity, the source layout of atmospheric nitrogen aerosol has changed. The inorganic nitrogen ($NH_4^+$ and $NO_3^-$) was essential part of atmospheric nitrogen aerosol and accounted for 69%. To comprehensively clarify the level, sources, and environmental fate of $NH_4^+$ and $NO_3^-$, their concentrations and stable isotopes ($\delta^{15}N$) in fine particulate matters ($PM_{2.5}$) were measured in a subtropical megacity of South China. $N-NH_4^+$ and $N-NO_3^-$ contributed 45.8% and 23.2% to total nitrogen (TN), respectively. The source contributions of $NH_4^+$ and $NO_3^-$ were estimated by $\delta^{15}N$, which suggested that anthropogenic combustion activities including coal combustion, biomass burning, and vehicles were dominant sources. Especially, biomass burning was the predominant source of $NH_4^+$ (27.9%). Whereas, coal combustion was the dominant source of $NO_3^-$ (40.4%). This study emphasized the substantial impacts of human activities on inorganic Nr. With the rapid development of industry and transportation, nitrogen emissions will be even higher. The promotion of clean energy and efficient use of biomass would help reduce nitrogen emissions and alleviate air pollution.

## 1. Introduction

Nitrogenous aerosols are ubiquitous in environment and play an important role as nutrients in ecosystems(Bhattarai et al., 2019). With the massive combustion of fossil fuels and the development of livestock, the proportion of TN in particulate matter (PM) ranges from 1.2% to 17.0% and has shown a rapid increase in the last few decades(Bhattarai et al., 2019; Galloway et al., 2004; Holland et al., 1999). Mostly nitrogenous aerosols formed from atmospheric Nr will be deposited into terrestrial and aquatic ecosystems(Huang et al., 2015). Excessive external nitrogen deposition accelerates nitrogen loss in soil, decreases species diversity, disturbs terrestrial ecosystems, and leads to eutrophication in aquatic ecosystems(Breemen, 2002; Wedin and Tilman, 1996; Yang et al., 2015). Furthermore, nitrogenous aerosols have adverse impacts on the climate, air quality, and human health(Bhattarai et al., 2019; Song et al., 2021).

$N-NO_3^-$ and $N-NH_4^+$ as inorganic Nr are dominant species in the deposition of nitrogen(Zhu et al., 2015). $N-NH_4^+$ was the highest in nitrogen deposition, and $NH_4^+$ was gradually considered to be an important component of secondary inorganic aerosols (SIA)(Sun et al., 2021). $NH_3$, the precursor of $NH_4^+$, is a vital atmospheric alkaline gas, which can participate in nucleation to promote new particles generation, and can react with acid gas to produce ammonium sulfate and ammonium nitrate(Dunne et al., 2016; Fu et al., 2017). The excessive $NH_3$ emission from anthropogenic sources will partially offset the benefits of reducing $SO_2$ and NOx and trigger urban haze in China(Sun et al., 2021; Meng et al., 2018; Pan et al., 2018a). In many urban environments, $NO_3^-$ has replaced sulfate as the component with the highest proportion in SIA. NOx, precursors of $NO_3^-$, are also closely related to the formation of atmospheric oxidants and exert important effects on atmospheric oxidation. In addition, $NH_4NO_3$ in PM plays an increasingly important role in promoting the formation of sulfate and organic matter, and has profound effect on the physical and chemical properties of PM(Liu et al., 2021; Liu et al., 2020; Hodas et al., 2014). Therefore, to mitigate nitrogen deposition and air pollution, the control of $NH_4^+$ ($NH_3$) and $NO_3^-$ (NOx) should not be neglected.

Considerable efforts have been made to comprehensively understand the budget of atmospheric $NH_4^+$ and $NO_3^-$. $\delta^{15}N$ is effective to quantify sources contribution of nitrogenous

species(Elliott et al., 2007). The anthropogenic combustion sources (combustion of coal, biomass, and gasoline) play a key role in the emission of $NO_3^-$ (NOx) in many regions of China suggested by $\delta^{15}N$(Zong et al., 2020), which also have large effects on $NH_3$(Chen et al., 2022b). $NH_3$ is released by agricultural sources (agricultural activity and livestock) and non-agricultural sources (fossil fuel combustion and vehicle)(Bhattarai et al., 2019). A previous study showed that agricultural source was the dominant source (80%-90%) of $NH_3$ in China(Kang et al., 2016). However, $NH_3$ emissions from agricultural source have been reduced due to intensive farming and efficient fertilization(Wang et al., 2022). The incomplete burning of biomass leads to massive $NH_3$ emissions and is gradually to be the second largest non-agricultural source of $NH_3$(Yu et al., 2020), which may be responsible for the lag of the decline in air pollutants deposition behind the reduction in emission of precursors(Zhao et al., 2022b). Biomass burning in the suburbs also has a potential impact on urban $NH_3$(Xiao et al., 2020). As for urban $NH_3$, combustion sources (including coal combustion, vehicles emission, and biomass burning) were gradually becoming dominant sources in recent years verified by $\delta^{15}N$-NHx ($NH_3$+$NH_4^+$)(Xiao et al., 2020; Pan et al., 2018b). In addition, the super clean emission of coal-fired power plant and strict emission standards of vehicles will change the source layout of $NH_4^+$ and $NO_3^-$. Selective catalytic reduction technology equipped with vehicles and industrial boiler reduces NOx but increases $NH_3$ emissions(Meng et al., 2017; Pan et al., 2016). The occurrence of haze in North China was closely related to $NH_3$ emissions from combustion sources(Pan et al., 2018a; Pan et al., 2018b). $NH_4^+$ and $NO_3^-$ are the main components of SIA and play a vital role in the formation of secondary aerosol(Meng et al., 2017), so it is necessary to revisit their sources.

Nr emissions from densely populated subtropical areas increased rapidly with the high development of industry and transportation(Wang et al., 2013). Guangzhou is the core megacity in the South subtropical region of China, where the atmospheric environment is complex and the atmospheric oxidation level is high(Tan et al., 2019). The high emissions of inorganic nitrogen from anthropogenic combustion sources have serious and profound impacts on the environment. In this study, we aimed to comprehensively clarify the level of inorganic Nr and revisit the source layout of atmospheric inorganic Nr.

## 2. Experimental and theoretical methods

### 2.1. Sampling and Chemical concentration analysis

PM$_{2.5}$ samples (n=66) were collected from May 2017 to June 2018 in Guangzhou (23.13°N, 113.27°E). Details of sample collection can be found in our previous study(Jiang et al., 2021a). The chemical components including water-soluble ions (i.e., NH$_4^+$, K$^+$, Na$^+$, Ca$^{2+}$, Mg$^{2+}$, Cl$^-$, NO$_3^-$, and SO$_4^{2-}$), organic carbon (OC), element carbon (EC), and organic molecular markers (e.g., levoglucosan) were analyzed in our previous studies (**SI Text S1**)(Jiang et al., 2021a; Jiang et al., 2021b). Moreover, meteorological parameters (temperature, relative humidity (RH), atmosphere pressure, and wind speed) and the concentration of trace gases (CO, SO$_2$, NO, NO$_2$, and O$_3$) were acquired by online instruments (details shown in **SI Text S1**). A circular punch (r=1cm) of the sample filter was wrapped in a tin boat and then measured in an elemental analyzer to determine the concentrations of TN.

### 2.2. Isotope analysis

The $\delta^{15}$N-NO$_3^-$ and $\delta^{18}$O-NO$_3^-$ values in PM$_{2.5}$ were analyzed by methods of nitrous oxide (N$_2$O), which was described in previous study in detail(Zong et al., 2017). Briefly, NO$_3^-$ was reduced to NO$_2^-$ using cadmium powder and imidazole solution, and N$_2$O was made by adding NaN$_3$ to NO$_2^-$ solution. The production of 75nmol N$_2$O gas was needed to measure. The N$_2$O gas produced by above processes was measured by MAT253 stable isotope mass spectrometer. The values of $\delta^{18}$O and $\delta^{15}$N were expressed in per mil (‰) shown in Eq. (1) and (2), relative to the international oxygen and nitrogen isotope standard, respectively.

$$\delta^{15}\text{N} = \left[ \frac{(^{15}\text{N}/^{14}\text{N})_{sample}}{(^{15}\text{N}/^{14}\text{N})_{standard}} - 1 \right] * 1000 \qquad (1)$$

$$\delta^{18}\text{O} = \left[ \frac{(^{18}\text{O}/^{16}\text{O})_{sample}}{(^{18}\text{O}/^{16}\text{O})_{standard}} - 1 \right] * 1000 \qquad (2)$$

The $\delta^{15}$N-NH$_4^+$ was measured by methods of hypobromite oxidation coupled with reduction of hydroxylamine hydrochloride(Sun et al., 2021). Briefly, NH$_4^+$ was oxidized to NO$_2^-$ using alkaline hypobromite (BrO$^-$), and N$_2$O was made by adding sodium arsenite and hydrochloric acid to NO$_2^-$ solution. The production of 120 nmol N$_2$O gas was needed to measure. The N$_2$O gas produced by above processes was measured by MAT253 stable isotope mass spectrometer. The values of $\delta^{15}$N were expressed in per mil (‰), Eq. (1). To ensure the

stability of the instrument, standard samples were tested for every ten samples. The standard deviation of replicates was generally less than 0.4‰, 0.8‰, and 0.5‰ for $\delta^{15}$N-NO$_3^-$, $\delta^{18}$O-NO$_3^-$, and $\delta^{15}$N-NH$_4^+$, respectively. The instrumental values of $\delta^{15}$N-NO$_3^-$ and $\delta^{18}$O-NO$_3^-$ were corrected by multi-point correction ($\delta^{18}$O $r^2$=0.99, $\delta^{15}$N $r^2$=0.999) based on international standards (IAEA-NO-3, USGS32, USGS34, and USGS35). The measured values of $\delta^{15}$N-NH$_4^+$ were also corrected by multi-point correction ($r^2$=0.999) based on international standards (IAEA-N1, USGS25, and USGS26). In addition, [7]Be and [210]Pb were acquired and details were shown in **SI Text S1**.

### 2.3. IsoSource and Bayesian mixing model

**IsoSource model.** IsoSource model was released by Environmental Protection Agency (EPA), could calculate ranges of source contributions to a mixture based on conservation of isotopic mass when number of sources is too large to permit a unique solution and provide the distribution of source proportions (Phillips et al., 2005). IsoSource model coupled with $\delta^{15}$N-NH$_3$ of atmospheric initial and potential sources (shown in **Table 1**) were applied to quantify the contribution of various sources to NH$_3$. Nitrogen fertilizers application, livestock, human waste, biomass burning, coal combustion, and vehicles were considered as sources of NH$_3$ in this study, details shown in **SI Text S2**. Atmospheric initial $\delta^{15}$N-NH$_3$ was calculated by following Eq. (3).

$$\delta^{15}\text{N-NH}_{3\text{-initial}} = \delta^{15}\text{N-NH}_4^+ - \varepsilon(\text{NH}_4^+\text{-NH}_3) \times (1-f) \qquad (3)$$

Where, $\delta^{15}$N-NH$_4^+$ and $\delta^{15}$N-NH$_{3\text{-initial}}$ represent the $\delta^{15}$N of particulate NH$_4^+$ and atmospheric initial NH$_3$, respectively. $\varepsilon$(NH$_4^+$-NH$_3$) represents the isotope fractionation factor in the gaseous NH$_3$ conversion to particulate NH$_4^+$ in the atmosphere. The f value represents the proportion of the initial NH$_3$ converted to NH$_4^+$, referring to NH$_3$ and NH$_4^+$ observed in Guangzhou (Liao et al., 2014).

The $\varepsilon$(NH$_4^+$-NH$_3$) value is temperature dependent(Huang et al., 2019), which can be deduced from(Urey, 1947), as shown in Eq. (4). The atmospheric average temperature was 24.5℃ in our sampling period, and the corresponding $\varepsilon$(NH$_4^+$-NH$_3$) value was 34.2‰ calculated by Eq. (4). In addition, the $\varepsilon$(NH$_4^+$-NH$_3$) in Guangzhou was estimated to be 32.4‰ according to Eq. (8). Eq. (8) was deduced by Eq. (5-7). According to Eq. (8), a linear fitting

equation was observed between $f$NH$_4^+$ and $\delta^{15}$N-NH$_4^+$ (**Fig. S1**), and the absolute value of the
slope (32.4‰) was equal to $\varepsilon$(NH$_4^+$-NH$_3$). The $\varepsilon$(NH$_4^+$-NH$_3$) average of the two methods (34.2‰
and 32.4‰) was 33.3‰ and approximated to the experimental isotope enrichment factor
(33‰)(Heaton et al., 1997). Therefore, 33‰ was used for deducing the $\delta^{15}$N of the initial NH$_3$.
$\varepsilon_{(\text{NH}_4^+\_\text{NH}_3)} = 12.4678 * \frac{1000}{T+273.15} - 7.6694$        (4)
$\delta^{15}\text{N-NH}_4^+ - \delta^{15}\text{N-NH}_3 = \varepsilon_{(\text{NH}_4^+\_\text{NH}_3)}$          (5)
$f\text{NH}_4^+ + f\text{NH}_3 = 1$            (6)
$\delta^{15}\text{N-NH}_4^+ * f\text{NH}_4^+ + \left(\delta^{15}\text{N-NH}_4^+ - \varepsilon_{(\text{NH}_4^+\_\text{NH}_3)}\right) * (1 - f\text{NH}_4^+) = \delta^{15}\text{N}$  (7)
$\delta^{15}\text{N-NH}_4^+ = -\varepsilon_{(\text{NH}_4^+\_\text{NH}_3)} * f\text{NH}_4^+ + \left(\delta^{15}\text{N} + \varepsilon_{(\text{NH}_4^+\_\text{NH}_3)}\right)$        (8)
Where, T represents the atmospheric temperature (°C). $\delta^{15}$N-NH$_4^+$ and $\delta^{15}$N-NH$_3$
represent the $\delta^{15}$N of particulate NH$_4^+$ and atmospheric NH$_3$, respectively. $\delta^{15}$N represents the
sum of $\delta^{15}$N-NH$_4^+$ and $\delta^{15}$N-NH$_3$. fNH$_3$ and fNH$_4^+$ represent the proportion of atmospheric
NH$_3$ and particulate NH$_4^+$, respectively.
**Bayesian mixing model.** $\delta^{15}$N were used for tracing source based on conservation of
isotopic mass. Bayesian mixing model improved upon linear mixing models by explicitly
considering uncertainty in prior information and isotopic equilibrium fractionation. Recently,
Bayesian mixing model was applied to trace the sources of atmospheric pollutants(Zong et al.,
2017; Zong et al., 2020). The model coupled with $\delta^{15}$N-NO$_3^-$ and $\delta^{18}$O-NO$_3^-$ were used to
identify the formation process and quantify the sources contribution of NO$_3^-$.
In Central Pearl River Delta (PRD), NO$_3^-$ formed through ·OH and N$_2$O$_5$ pathways
contributed to 94% simulated by CAMQ model (Qu et al., 2021). In this study, only ·OH and
N$_2$O$_5$ formation pathways were considered. Details of NO$_3^-$ formation pathway were also
shown in **SI Text S2**. The atmospheric $\delta^{18}$O-NO$_3^-$ can be expressed by Eq. (9). The [$\delta^{18}$O-
HNO$_3$]$_\text{OH}$ can be further expressed by Eq. (10) assuming no kinetic isotope fractionation
(Walters and Michalski, 2016). And [$\delta^{18}$O-HNO$_3$]$_\text{H}_2\text{O}$ can be estimated by Eq. (11) (Walters and
Michalski, 2016). The $\delta^{18}$O values in tropospheric H$_2$O, NOx, O$_3$, and OH were within a certain
range. The tropospheric $\delta^{18}$O-H$_2$O, $\delta^{18}$O-NOx, $\delta^{18}$O-O$_3$, and $\delta^{18}$O-OH ranged from -25‰ to
0‰(Baskaran et al., 2011; Walters and Michalski, 2016), 112‰ to 122‰ (Michalski et al.,
2014; Walters and Michalski, 2016), 90‰ to 122‰, and -15‰ to 0‰, respectively(Fang et al.,
2011; Johnston and Thiemens, 1997). Therefore, the $\gamma$ (the contribution of ·OH formation
pathway) can be estimated by $f$NO$_2$ and oxygen isotope fractionation i.e., $\alpha$NO$_2$/NO, $\alpha$OH/H$_2$O,
and $\alpha$N$_2$O$_5$/NO$_2$. The oxygen isotope fractionations are temperature dependent and can be
estimated by Eq. (13) and **Table S1.** The $f$NO$_2$ varied from 0.20 to 0.95(Zong et al., 2017;
Walters et al., 2016). Based on $\delta^{18}$O-NO$_3^-$, $\delta^{18}$O-H$_2$O, $\delta^{18}$O-NOx, $\delta^{18}$O-O$_3$, and temperature
(Eq. (9-13)), $\gamma$ (maximum $\gamma$ and minimum $\gamma$) was estimated by Monte Carlo simulation nested
in Bayesian mixing model (Zong et al., 2017). Assuming no kinetic isotope fractionation, the
nitrogen isotope fractionation value in the formation process of NO$_3^-$ ($\varepsilon$N) was calculated by
Eq. (13-16) combined with $\gamma$ and temperature (Zong et al., 2017; Walters and Michalski, 2016;
Walters et al., 2016). The $\varepsilon$N value in our sampling period was 5.1±2.5‰, which was
comparable to that in Beijing(average 6.5‰)(Fan et al., 2020). The contributions of different
sources to atmospheric NOx were quantified by Bayesian mixing model coupled with $\varepsilon$N, $\delta^{15}$N-
atmospheric-NO$_3^-$, and $\delta^{15}$N-NOx endmembers shown in **Table 1**. We considered coal
combustion, mobile traffic sources, biomass burning, and soil microbial process as dominant
atmospheric NOx sources in Guangzhou, details shown in **SI Text S2**. The specific details of
Bayesian mixing model were reported by our previous studies(Zong et al., 2017; Zong et al.,

189    2020).

$\delta^{18}$O-NO$_3^-$ = $\gamma \times [\delta^{18}$O-NO$_3^-]_{OH}$ + $(1-\gamma) \times [\delta^{18}$O-NO$_3^-]_{H_2O}$ = $\gamma \times [\delta^{18}$O-HNO$_3]_{OH}$ +
$(1-\gamma) \times [\delta^{18}$O-HNO$_3]_{H_2O}$ $\qquad\qquad\qquad\qquad\qquad\qquad$ (9)
$[\delta^{18}$O-HNO$_3]_{OH}$ = $\frac{2}{3}[(\delta^{18}$O-NO$_2)]_{OH}$ + $\frac{1}{3}[\delta^{18}$O-OH$]_{OH}$ = $\frac{2}{3}\left[\frac{1000\times(^{18}\alpha_{NO_2/NO}-1)(1-f_{NO_2})}{(1-f_{NO_2})+(^{18}\alpha_{NO_2/NO}\times f_{NO_2})}\right.$ +
$\left. [\delta^{18}$O-NO$_X]\right]$ + $\frac{1}{3}[(\delta^{18}$O-H$_2$O) + $1000 \times (^{18}\alpha_{OH/H_2O}-1)]$ $\qquad\qquad$ (10)
$[\delta^{18}$O-HNO$_3]_{H_2O}$ = $\frac{5}{6}(\delta^{18}$O-N$_2$O$_5)$ + $\frac{1}{6}(\delta^{18}$O-H$_2$O) $\qquad\qquad\qquad$ (11)
$\delta^{18}$O-N$_2$O$_5$ = $\delta^{18}$O-NO$_2$ + $1000 \times (^{18}\alpha_{N_2O_5/NO_2}-1)$ $\qquad\qquad\qquad$ (12)
$1000(^m\alpha_{X/Y}-1)$ = $\frac{A}{T^4} \times 10^{10}$ + $\frac{B}{T^3} \times 10^8$ + $\frac{C}{T^2} \times 10^6$ + $\frac{D}{T} \times 10^4$ $\qquad$ (13)
$\varepsilon$N = $\gamma \times \varepsilon(\delta^{15}$N-NO$_3^-)_{OH}$ + $(1-\gamma) \times \varepsilon(\delta^{15}$N-NO$_3^-)_{H_2O}$
$$= \gamma \times \varepsilon(\delta^{15}\text{N-HNO}_3)_{\text{OH}} + (1 - \gamma) \times \varepsilon(\delta^{15}\text{N-HNO}_3)_{\text{H}_2\text{O}} \qquad (14)$$
$$\varepsilon(\delta^{15}\text{N-HNO}_3)_{\text{OH}} = \varepsilon(\delta^{15}\text{N-NO}_2)_{\text{OH}} = 1000 \times \left[\frac{(^{15}\alpha_{\text{NO}_2/\text{NO}}-1)(1-f_{\text{NO}_2})}{(1-f_{\text{NO}_2})+(^{15}\alpha_{\text{NO}_2/\text{NO}} \times f_{\text{NO}_2})}\right] \quad (15)$$
$$\varepsilon(\delta^{15}\text{N-HNO}_3)_{\text{H}_2\text{O}} = \varepsilon(\delta^{15}\text{N-N}_2\text{O}_5)_{\text{H}_2\text{O}} = 1000 \times \left(^{15}\alpha_{\text{N}_2\text{O}_5/\text{NO}_2} - 1\right) \qquad (16)$$
Where, $\gamma$ is the contribution of ·OH formation pathway to $NO_3^-$, $\varepsilon N$ is the nitrogen isotope
fractionation value. $fNO_2$ is the fraction of $NO_2$ in the total NOx. $^{18}\alpha NO_2/NO$, $^{18}\alpha OH/H_2O$,
$^{18}\alpha N_2O_5/NO_2$ are the oxygen isotope equilibrium fractionation factors between $NO_2$ and
NO, ·OH and $H_2O$, $N_2O_5$ and $NO_2$, respectively. $^{15}\alpha NO_2/NO$ and $^{15}\alpha N_2O_5/NO_2$ are the nitrogen
isotope equilibrium fractionation factor between $NO_2$ and NO, $N_2O_5$ and $NO_2$, respectively.
**Table 1.** The estimation of $\delta^{15}$N-NH$_3$ and $\delta^{15}$N-NOx from various sources.

| Source | $\delta^{15}$N-NH$_3$(‰) | References |
|---|---|---|
| Biomass burning | 17.5±7.8 | (Kawashima and Kurahashi, 2011; Xiao et al., 2020) |
| Coal combustion | -2.5±6.4 | (Felix et al., 2013; Pan et al., 2016) |
| Urban traffic | 6.6±2.1 | (Walters et al., 2020) |
| Fertilizer | -28.3±5.8 | (Bhattarai et al., 2021; Chang et al., 2016; Felix et al., 2013; Bhattarai et al., 2020) |
| Livestock | -18.3±7.7 | (Bhattarai et al., 2021; Chang et al., 2016; Felix et al., 2013; Bhattarai et al., 2020) |
| Urban waste | -22.8±3.6 | (Bhattarai et al., 2021; Chang et al., 2016) |
| Source | $\delta^{15}$N-NOx(‰) | References |
| Biomass burning | 1.04±4.13 | (Zong et al., 2017; Fibiger and Hastings, 2016; Zong et al., 2022) |
| Coal combustion | 13.72±4.57 | (Zong et al., 2017; Felix et al., 2015; Felix et al., 2012) |
| Mobile source | -7.25±7.80 | (Zong et al., 2017; Walters et al., 2015) |
| Soil microbial process | -33.77±12.16 | (Zong et al., 2017; Felix and Elliott, 2013) |

## 3.  Results and discussion

### 3.1. Concentration and seasonal variation of NH$_4^+$ and NO$_3^-$

The concentration of $NH_4^+$ and $NO_3^-$ in PM$_{2.5}$ was 1.6±1.3 μg m$^{-3}$ and 2.8±3.4 μg m$^{-3}$,
contributed 18.7% and 32.6% to SIA. The concentration of N-NH$_4^+$ and N-NO$_3^-$ was 1.2±1.0
μg m$^{-3}$ and 0.6±0.8 μg m$^{-3}$, contributed 45.8% and 23.2% to TN, respectively; thus, NH$_4^+$ and
NO$_3^-$ were essential part of nitrogen aerosols. NH$_4^+$ and NO$_3^-$ showed similar seasonal
variations with higher concentrations in winter than in summer (**Fig. 1**). During winter the air
mass was often dry and cold with low wind speed, which meant the decrease of the atmospheric
self-purification capability. In addition, primary combustion sources related to fossil fuel and
biomass burning always showed significant increase in North China in winter, which greatly

increased the concentration of atmospheric pollutants in Guangzhou by long-range transportation. However, during summer, the air mass from sea was relatively clean with high wind speed facilitating the diffusion of pollutants. Moreover, high temperature in summer was conducive to the decomposition of $NH_4NO_3$(Song et al., 2008). Thus, the levels of $NH_4^+$ and $NO_3^-$ were lower in summer. In addition, concentrations of $NH_4^+$ and $NO_3^-$ in our study, were lower than North China [Beijing(Wu et al., 2019; Fan et al., 2022), Tianjin(Xiang et al., 2022), Shijiazhuang(Xiang et al., 2022), and Harbin(Sun et al., 2021)], East China [Nanchang(Xiao et al., 2020)], and Central China [Wuhan and Changsha(Xiao et al., 2020; Zong et al., 2020)], suggested the level of air pollution in Guangzhou has been alleviated to a certain extent. Therefore, it is necessary to conduct comprehensive study on the emission sources of $NH_4^+$ and $NO_3^-$ to take more effective measures to mitigate air pollution.

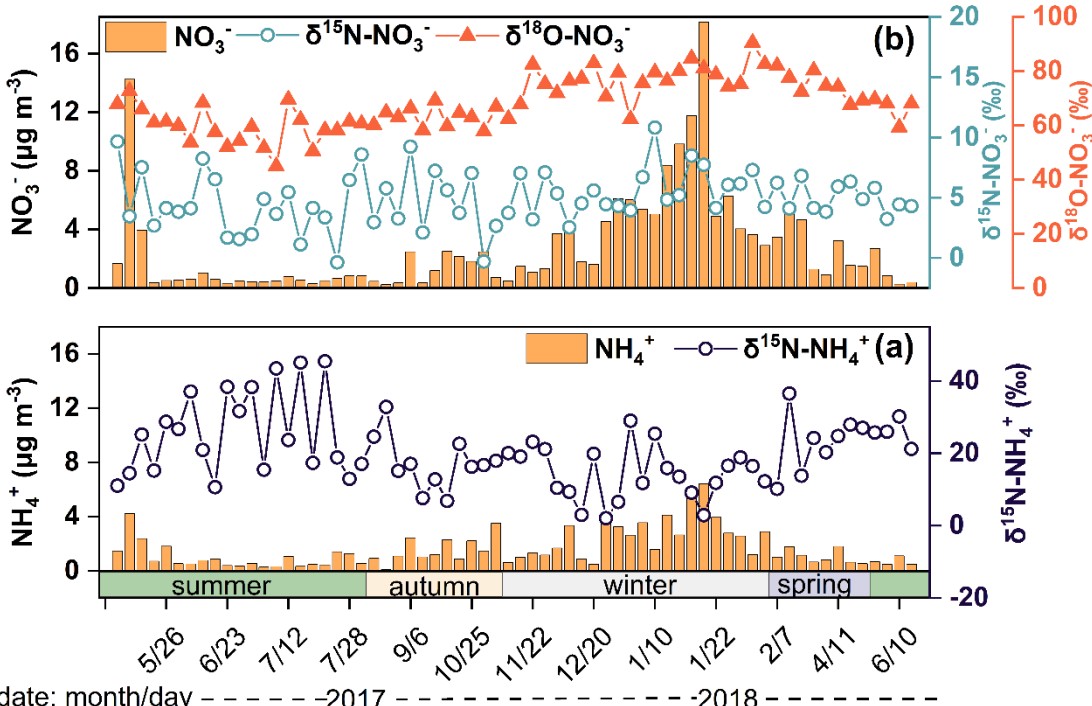

**Figure 1.** The concentration and $\delta^{15}N$ of $NH_4^+$ (a) and concentration, $\delta^{15}N$, and $\delta^{18}O$ of $NO_3^-$ (b).

**3.2. Characteristic and seasonal variation in $\delta^{15}N$-$NH_4^+$ and source apportionment of $NH_4^+$**

The $\delta^{15}N$-$NH_4^+$ values over Guangzhou ranged from 2.1‰ to 45.5‰, with an annual mean of 20.2±10.1‰. In our study, the $\delta^{15}N$-$NH_4^+$ values were comparable to those at suburban sites (**Fig. S2**) such as sites in Japan (22.1±8.3‰, 16.1±6.6‰)(Kawashima and Kurahashi, 2011) and Korea (Jeju Island,17.4±4.9‰)(Kundu et al., 2010) but heavier than those in polluted regions, such as Guangzhou during summer haze(average 7.17‰)(Liu et al., 2018) and Beijing (-37.1‰ to 5.8‰)(Pan et al., 2016). $\delta^{15}N$-$NH_4^+$ values were lower in autumn (17.3‰) and winter (14.4‰) than in spring (22.5‰) and summer (25.7‰), which was similar to the trends in Japan(Kawashima and Kurahashi, 2011).

The seasonal differences in $\delta^{15}N$-$NH_4^+$ values were significant between warm (summer/spring) and cold seasons (winter/ autumn) ($p < 0.05$). The $\delta^{15}N$-$NH_4^+$ was affected by the ratio of $NH_4^+/(NH_3+NH_4^+)$ (Eq. (8) and **Fig. S1**). A linear fitting equation was observed between $NH_4^+/(NH_3+NH_4^+)$ and $\delta^{15}N$-$NH_4^+$, and the absolute value of the slope (32.4) approximated the isotope equilibrium fractionation value (33%) between atmospheric $NH_3$ and $NH_4^+$ (**Fig. S1**). The linear fitting suggested that the lower the $NH_4^+$ proportion was, the heavier the $\delta^{15}N$-$NH_4^+$ value. The lower $NH_4^+$ level was accordance with higher $\delta^{15}N$-$NH_4^+$ in summer, which was the opposite of winter. In addition, previous study suggested that the marked variation in $\delta^{15}N$-$NH_4^+$ values was largely controlled by the emission sources of $NH_3$, the precursor gas of $NH_4^+$(Liu et al., 2018). According to the $\delta^{15}N$-$NH_4^+$ results, the source of $NH_4^+$ was assigned as biomass burning (27.9±16.4%), coal combustion (16.0±3.9%), vehicles (19.8±5.3%), fertilizer (10.9±6.1%), livestock (12.7±5.8%), and urban waste (11.9±6.1%), shown in **Fig. 2a**.

In our study, non-agriculture sources were the dominators of $NH_4^+$ (75.6%). Unexpectedly, the contribution of biomass burning was the highest. Especially, from late June to July, the contribution of biomass burning enhanced, which possibly resulted from sugarcane leaf burning. The $\delta^{15}N$ in sugarcane leaf was as high as 38‰(Martinellia et al., 2002). The $\delta^{15}N$ of $NH_4^+$ formed from $NH_3$ released by sugarcane leaves burning was 44.1‰ (**SI Text S3**), which was consistent with the highest $\delta^{15}N$-$NH_4^+$ values (45.5‰ and 45.1‰) in July. In PRD, south winds prevail in July and the sampling site is located downwind of sugarcane planting area.

Therefore, the air mass to the sampling site might carry the pollutants related to sugarcane leaf
burning. $K^+$ is a typical biomass burning tracer(Cui et al., 2018). Considering the impact of
primary emission intensity, $[NH_4^+/EC]$ and $[K^+/EC]$ were used to calculate the correlation
coefficient (r=0.435, $p < 0.01$), which verified $NH_4^+$ was influenced by biomass burning. In
recent years, biomass burning has been gradually identified as an important source of
$NH_4^+$(Meng et al., 2017; Xiao et al., 2020). The results based on emission inventories showed
that the contribution of residential biomass combustion to $NH_3$ ranged from 33% to 53% in
China(Meng et al., 2017). According to $\delta^{15}N$, biomass burning contributed 18% [Harbin, East
North China](Sun et al., 2021), 46%[Wuhan, South Central China], 40% [Changsha, South
Central China](Xiao et al., 2020), 35% [Nanchang, East China](Xiao et al., 2020), and 23%
[Guangzhou, South China](Chen et al., 2022a) to $NH_4^+$. Particularly, in Guangzhou the
contribution of biomass burning in the ground was higher than that in Guangzhou tower with
a height of 488 meters, suggested that the influence of regional biomass burning(Chen et al.,
2022a). Furthermore, $^7Be$ mainly originates from upper atmosphere, whereas $^{210}Pb$ is derived
from terrestrial surface(Jiang et al., 2021b). High level of $^7Be$ observed in ground suggested
the sink influence of upper atmosphere. $^7Be$ and $^{210}Pb$ are chemically stable and with unique
sources, which can effectively reflect the transport of continental air mass and the air exchange
between stratosphere and troposphere. In our study, the correlation coefficient between $NH_4^+$
and $^{210}Pb$ (r=0.701, $p < 0.01$) was higher than that between $NH_4^+$ and $^7Be$ (r=0.432, $p < 0.01$),
suggested that $NH_4^+$ was mainly affected by regional emission. Therefore, biomass burning
exerted essential influence on $NH_4^+$ level, which should no longer be ignored.

In addition, with the acceleration of urbanization, combustion sources related to fossil

fuels have become the main sources of $NH_3$. In previous studies, the source of $NH_x$ ($NH_3+NH_4^+$)
was mainly from agricultural activity due to rough way of farming(Chang et al., 2016; Pan et
al., 2020). However, with the improvement of efficient fertilization practices, agricultural $NH_3$
decreased significantly(Wang et al., 2022). Fossil fuels, such as coal and gasoline, are major
energies for production and domestic using, and their contribution to $NH_3$ has become
increasingly important. In North China, fossil fuel combustion contributed 92% to $NH_3$ during
hazes(Zhang et al., 2020; Pan et al., 2016). In previous study of Guangzhou, the contribution
of $NH_3$ from fossil source in ground observations (43%) was higher than the observed in
Guangzhou tower (18%), indicated the importance of locally related fossil fuel combustion
source(Chen et al., 2022a). In our study, vehicle emission and coal combustion contributed
19.8±5.3% and 16.0±3.9% of $NH_4^+$ respectively, which was lower than North China but higher
than agricultural sources. The share of $NH_3$ from vehicle exhaust was estimated to be 18.8%
based on the emission factor of $NH_3$ from on road vehicles in Guangzhou, which was similar
to our results(Liu et al., 2014). The selective catalytic reduction process for vehicle can reduce
NOx, but increased emission of $NH_3$, which has confirmed as an important source of $NH_3$(Heeb
et al., 2006; Meng et al., 2017). Despite the efforts of government to promote electric vehicles
in recent years, their share is still relatively low (about 5%). As increasing car ownership, this
has an important impact on atmospheric $NH_3$. Coal combustion was the second most important
source of fossil combustion after vehicle emissions in our study, although the contribution was
lower than in North China(Wu et al., 2019; Zhang et al., 2020; Pan et al., 2016). The absence
of heating in Guangzhou may explain the lower contribution of coal combustion compared to
the North. On an annual basis, the contribution of fossil fuel-related combustion sources in our
study (35.8%) was comparable to that in North China (37%-52%)(Pan et al., 2018a).
The source contributions of $NH_4^+$ in our study were compared to other regions, shown in
**Fig. S3**. The combustion related sources (biomass burning, coal combustion, and vehicle) have
gradually become the dominant source of urban atmospheric $NH_3$. Biomass burning and
vehicle could emit massive carbon monoxide (CO)(Li and Wang, 2007; Wang et al., 2005). In
Guangzhou, $NH_4^+$ was positively related to CO ($r$=0.637, $p < 0.01$), which confirmed
combustion sources played a key role in $NH_4^+$. From a historical perspective, $NH_3$ emissions
from anthropogenic combustion and industry have been steadily increasing since 1960(Meng
et al., 2017). The optimization of energy structure and encouragement of the development of
new energy vehicle would be hopeful to reduce $NH_3$. The results of this study would be
conducive to reducing $NH_3$ scientifically and effectively and would relieve the pressure on the
reduction from agricultural source.

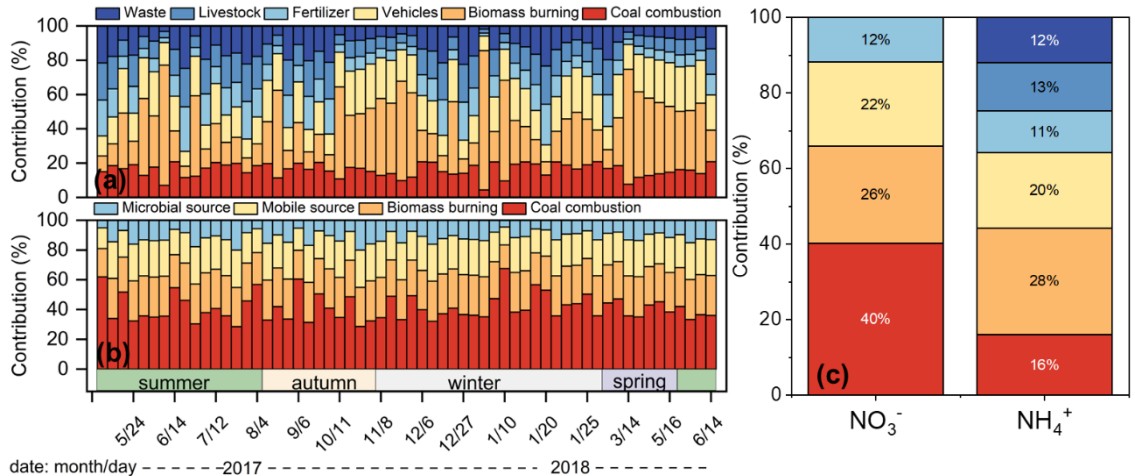


**Figure 2.** The sources apportionment results of atmospheric $NH_4^+$ (a) and $NO_3^-$ (b) in
Guangzhou, and the comparison of sources results between $NH_4^+$ and $NO_3^-$ (c).

### 3.3. Characteristic and seasonal variation in $\delta^{18}O\text{-}NO_3^-$ and $\delta^{15}N\text{-}NO_3^-$ and source apportionment of $NO_3^-$

#### 3.3.1. Seasonal variation of $\delta^{18}O\text{-}NO_3^-$

The $\delta^{18}O\text{-}NO_3^-$ in Guangzhou was 68.1±9.7‰ (44.9‰ to 90.5‰) comparable to that in
precipitation (66.3‰, ranging from 33.4‰ to 86.2‰)(Fang et al., 2011), but lower than those
regions with weak light intensity, such as Beihuangcheng Island (ranging from 49.4‰ to
103.9‰)(Zong et al., 2017) and Bermuda Islands (cold season 76.9±6.3‰) (Hastings et al.,
2003). In this study, $\delta^{18}O\text{-}NO_3^-$ was higher in winter and spring than in summer and autumn,
which was similar to the seasonal variation in $\delta^{18}O\text{-}NO_3^-$ in previous studies (Fang et al., 2011;
Gobel et al., 2013). On the one hand, $\delta^{18}O\text{-}NO_3^-$ value was associated with the formation
pathways of $NO_3^-$. The results simulated by Bayesian mixing model suggested that the
contributions of $N_2O_5$ channel to $NO_3^-$ were 56.8%, 58.9%, 29.2%, and 27.0% in winter, spring,
autumn, and summer, respectively. The $\delta^{18}O$ value of $NO_3^-$ formed by $N_2O_5$ channel is higher
than that by ·OH pathway (**SI Text S2**). The night in cold season was longer than that in warm
season, which favored $NO_3^-$ formation through $N_2O_5$ channel. In addition, the illumination
intensity was weakened in cold season compared with that in warm season, which constrained
the production of ·OH(Zong et al., 2020; Tan et al., 2019; Wang et al., 2017). Thus, the
contribution of the $N_2O_5$ channel in cold season was higher than that in warm season.
Furthermore, concentration of $NO_3^-$ was high when contribution of $N_2O_5$ channel enhanced
(**Fig. 3**), suggested $NO_3^-$ pollution was related to $N_2O_5$ hydrolysis pathway. The air mass to
Guangzhou was derived from the South China Sea in summer and the North continental region
in winter. The higher $\delta^{18}O\text{-}NO_3^-$ and $NO_3^-$ concentration might be affected by long-range and
high-altitude transport from North China, which might carry abundant precursors. Massive
$NO_3^-$ could be formed by $N_2O_5$ hydrolysis at high altitude and transported to the ground. The
index of $f(^7Be,^{210}Pb)$ was expressed in **SI Text S1** and could reflect the influence of atmospheric
dynamic transport on aerosol pollutants(Jiang et al., 2021b). Generally, air masses with low
values of $f(^7Be,^{210}Pb)$ suggested that pollutants were associated with continental surface
emission, whereas high $f(^7Be,^{210}Pb)$ were influenced by long-range transport from upper air
masses. The contribution of $N_2O_5$ channel was positively correlated with $f(^7Be,^{210}Pb)$ (r=0.319,
$p < 0.05$), indicated the long-range transport influence of upper air mass on $N_2O_5$ channel. For
example, on 25 January 2018, the contribution of $N_2O_5$ channel (nitrate) was 81.1% (3.6 μg m⁻
³), when the upper air mass was from North China. However, on 7 July 2017, the $N_2O_5$ channel
(nitrate) contributed only 5.7% (0.5 μg m⁻³) corresponding to the air mass mainly from the
South China Sea transported at low-altitude (**Fig. S4**).

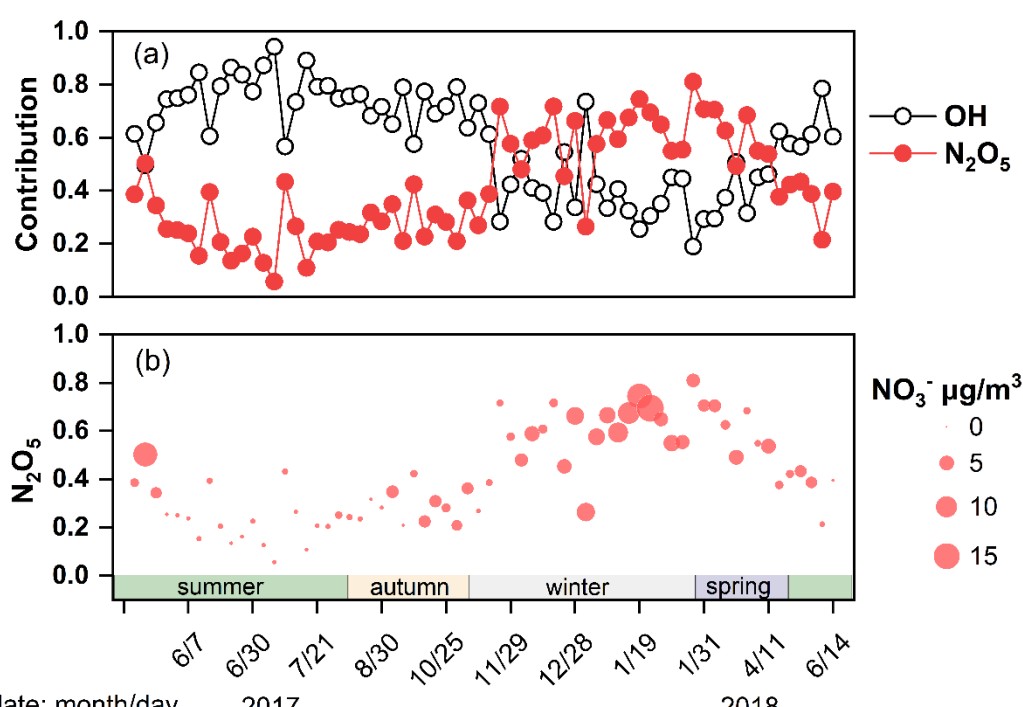


**Figure 3.** The contribution of the OH radical oxidation and $N_2O_5$ hydrolysis pathway to $NO_3^-$ (a). The vertical position of dots corresponded to the contribution of $N_2O_5$ pathway and the size of the dots corresponded to the concentration of $NO_3^-$ (b).

$\delta^{18}O$-$NO_3^-$ decreased from 76.7‰ in 2014 to 68.1‰ in 2017-2018(Zong et al., 2020), which indicated that ·OH channel became more important in Guangzhou. The enhanced contribution of ·OH pathway indicated the increasing atmospheric oxidation capacity. In recent years, although the concentration of $PM_{2.5}$ in Guangzhou has significantly decreased, the photochemical pollution caused by high $O_3$ concentrations was not optimistic(Tan et al., 2019). The $O_3$ concentration in the PRD showed a fluctuating upward trend from 2013 to 2020; especially in 2017-2018, $O_3$ concentrations were at high levels (Environmental Status Bulletin of Guangdong Province **Fig. S5**). In our study, the $NO_3^-$ formation pathway inferred from $\delta^{18}O$-$NO_3^-$ proved the enhancement of atmospheric oxidation capacity.

### 3.3.2. Seasonal variation of $\delta^{15}N$-$NO_3^-$ and source apportionment of $NO_3^-$

**Seasonal variation of $\delta^{15}N$-$NO_3^-$.** The $\delta^{15}N$-$NO_3^-$ in Guangzhou was 4.9±2.2‰ (-0.4‰ to 10.8‰), which was similar to the wet deposition(Fang et al., 2011). The $\delta^{15}N$-$NO_3^-$ was comparable to that from the Northeast United States (6.8‰)(Elliott et al., 2009), and lower than regions in China, where $NO_3^-$ was predominantly derived from anthropogenic sources, such as Heshan in Guangdong (7.50±3.30‰)(Su et al., 2020), Beihuangcheng Island (8.2±6.2‰)(Zong et al., 2017), and Beijing (12.1±3.3‰)(Fan et al., 2022). Nevertheless, the $\delta^{15}N$-$NO_3^-$ in this study was significantly higher than those from clean background regions, where $NO_3^-$ was mainly from natural sources, such as the coast of Antarctica (-12.0±15.6‰)(Savarino et al., 2007) and Bermuda (-2.1±1.5‰ warm season, -5.9±3.3‰ cold season)(Hastings et al., 2003). The values of $\delta^{15}N$-$NO_3^-$ in winter, spring, summer, and autumn were 5.6‰, 5.3‰, 4.4‰, and 4.5‰, respectively. The $\delta^{15}N$-$NO_3^-$ in winter and summer showed significant difference ($p < 0.05$). The values of $\delta^{15}N$-$NO_3^-$ were influenced by atmospheric processes and emission sources(Elliott et al., 2009). For $N_2O_5$ channel, $NO_3^-$ is characterized by higher $\delta^{15}N$ values(Freyer et al., 1993; Elliott et al., 2009). The $N_2O_5$ channel was the predominant formation pathway of $NO_3^-$ in winter, which was in accordance with the seasonal variation in $\delta^{15}N$-$NO_3^-$. In addition, the difference in $\delta^{15}N$-$NO_3^-$ reflected the variation

in the emission source of $NO_3^-$. $\delta^{15}N$-NOx from coal combustion was relatively high. In winter,
the higher $\delta^{15}N$-$NO_3^-$ was probably related to long-range transport from North, where coal
combustion enhanced in winter.
**Source apportionment of $NO_3^-$.** Based on the Bayesian mixing model coupled with $\delta^{15}N$-
$NO_3^-$, $NO_3^-$ sources were assigned as coal combustion 40.4±8.7%, biomass burning 25.6±2.1%,
mobile sources (vehicles) 22.3±3.1%, and microbial process 11.7±3.8%. **Figure 2b** and **Fig.**
**S6** showed the source contribution of $NO_3^-$ in Guangzhou and other regions in China,
respectively. Compared to earlier periods (2013-2014), the concentration of $NO_3^-$ from vehicle
and coal combustion decreased significantly(Zong et al., 2020), which resulted from the stricter
vehicle emission standard, promotion of new energy electric vehicles, and ultraclean
transformation of coal combustion(Guangdongprovince, 2014; Tang et al., 2019). However,
almost all production and domestic segments rely on energy generated from coal combustion,
which was still dominant source of $NO_3^-$ in 2017-2018. Coal combustion was affected not only
by local emissions but also by external air mass transmission. The contribution of coal
combustion was higher in winter than in summer, which probably related to the long-range
transportation from the North. Taking 10 January 2018 as an example, the contribution of coal
combustion sources to $NO_3^-$ was 67.5%, and the corresponding air mass was from the North
and transmitted to Guangzhou through high altitude. However, the air mass on 26 July 2017
was mainly from the South China Sea, which was transmitted through low-altitude to
Guangzhou. The contribution of coal burning to $NO_3^-$ on 26 July 2017 was 28.5% lower than
that on 10 January 2018.
As non-fossil combustion source, biomass burning was also an important source of $NO_3^-$
and accounted for 25.6%. The contribution of biomass burning and vehicle was stable
throughout a year. Generally, high intensity biomass burning occurred in winter in Guangdong
province (dry season, i.e., from November to March)(Xu et al., 2019). $K^+$ is a typical tracer of
biomass burning. The concentration of $K^+$ enhanced in winter ($0.4\mu g/m^3$) was higher than that
in summer ($0.2\mu g/m^3$) and autumn ($0.2\mu g/m^3$), respectively, indicating enhancement of
biomass burning intensity. Also, $NO_3^-$ concentration of biomass burning remarkably enhanced
in winter ($1.2\mu g/m^3$), and was higher than that in summer ($0.4\mu g/m^3$) and autumn ($0.3\mu g/m^3$),
respectively. However, coal combustion also enhanced in winter due to the demand for heating
in North China. Our sampling site was influenced by the air mass with high coal combustion
contribution from the North by long-range transportation, which may reduce the contribution
of biomass burning relatively. Thus, the contribution of biomass burning showed stable
compared with coal combustion. Another non-fossil source is related to soil microbial activity
and only contributed 11.7% to $NO_3^-$, which was unexpectedly lower than the results in earlier
periods (2013-2014). Generally, the microorganisms in soil emit NO through nitrification or
denitrification, which was affected by the amount of carbon and nitrogen nutrients in soil(Hall
and Matson, 1996). In earlier periods, due to the higher level of aerosols, the amount of
nutrients settling in soil was also higher, which was exemplified by the observation of dry and
wet deposition in Guangzhou(He et al., 2022; Zheng et al., 2020). In addition, the reduction of
cultivated land from 2013 to 2018 might also reduce the contribution of microbial source
emissions. Therefore, emissions from natural sources were also influenced by human activities
to some extent. The contribution of microbial process was higher in summer than in winter. In
summer, higher RH and temperature were favorable for the intense activity of soil
microorganisms(Zong et al., 2017). The contributions of microbial processes to $NO_3^-$ also
decreased in winter compared with summer at regional background sites and five Chinese
megacities, including Guangzhou(Zong et al., 2017; Zong et al., 2020).
The sources comparison between $NO_3^-$ and $NH_4^+$ was shown in **Fig. 2c**. Coal combustion,
biomass burning, and vehicles were three significant sources of $NO_3^-$ and $NH_4^+$. Coal
combustion and biomass burning were the dominant sources of $NO_3^-$ and $NH_4^+$, respectively.
The vehicles were also an important source of atmospheric inorganic Nr contributed to 22.3%
and 19.8% of $NO_3^-$ and $NH_4^+$, respectively. Recently, the government has actively taken many
measures to reduce the pollution from vehicles, such as stricter automobile emission standards
and the promotion of new energy vehicles. However, due to the large vehicle ownership base,
the pollutants emitted from vehicles are not optimistic. In addition, vehicles emissions could
contribute half of the fresh secondary organic aerosol in urban environment(Zhang et al., 2022;
Zhao et al., 2022a).

## 4. Conclusions

A year-long field observation was conducted in Guangzhou to clarify the atmospheric fate of inorganic nitrogen aerosol. Inorganic nitrogen species were the most essential component of TN including $NH_4^+$ (45.8%) and $NO_3^-$ (23.2%), which are also dominant components of SIA and play a key role in China haze. The $\delta^{15}N$ is a powerful tool to quantify the source contribution of $NH_4^+$ and $NO_3^-$, which suggested that anthropogenic combustion sources (coal combustion, biomass burning, and vehicles) were the dominant sources.

Anthropogenic combustion sources contributed 63.2% to $NH_4^+$ higher than agricultural sources (23.6%). $NH_3$ largely facilitates the formation of sulfate and nitrate. Meanwhile, sulfate and nitrate promote each other with positive feedback effect, which could trigger haze. In megacities of China, the focus of $NH_3$ reduction should be on anthropogenic combustion sources, especially on biomass burning, which might be responsible for the lag of the decline in the deposition of air pollutions behind the reduction in emission(Zhao et al., 2022b). In addition, anthropogenic combustion sources accounted for 88.3% of $NO_3^-$. Coal combustion and vehicles contributed 40.4% and 22.3% to $NO_3^-$, respectively. Despite a series of measures to reduce emissions of NOx, fossil fuels, as the main energy for production and living, will still inevitably emit a large amount of NOx. Our results emphasized that the emission of atmospheric inorganic nitrogen is largely related to anthropogenic combustion sources. The development and promotion of clean energy and efficient use of biomass are conducive to the deep reduction of atmospheric nitrogen.

**Data availability**

The original data of this research (stable nitrogen isotopes and inorganic nitrogen concentrations) are available at Mendeley data (Li and Li, 2023). The Iso Source model was downloaded from Environmental Protection Agency, via their website: https://www.epa.gov/sites/default/files/2015-11/isosourcev1_3_1.zip.

**Author contributions**

Funding acquisition: Jun Li

Investigation: Tingting Li, Zeyu Sun, and Hongxing Jiang
Methodology: Tingting Li, Zeyu Sun, Hongxing Jiang, Jun Li, and Chongguo Tian
Project Administration: Jun Li
Resources: Jun Li, Chongguo Tian, and Gan Zhang
Software: Tingting Li, Zeyu Sun, and Chongguo Tian
Validation: Tingting Li and Jun Li
Writing – original draft: Tingting Li
Writing – review & editing: Jun Li
**Competing interests**
The authors declare that they have no conflict of interest.
**Financial support**
This study was supported by the Natural Science Foundation of China (NSFC; Nos.
(41977177), Guangdong Basic and Applied Basic Research Foundation (2021A1515011456),
Guangdong Foundation for Program of Science and Technology Research (Grant No.
2019B121205006 and 2020B1212060053).

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
