# Peer review of "High contribution of anthropogenic combustion sources to atmospheric inorganic reactive nitrogen in South China evidenced by isotopes"

_Atmospheric Chemistry and Physics, 2023_

## Author Comment (AC2)

**Response to Referee #2**

RC- Reviewer's Comments; AC – Authors' Response Comments

RC1: This paper estimated the relative contributions of main sources to ammonium and nitrate aerosols in a subtropical megacity of South China using stable N isotope analysis. They found that anthropogenic activities (e.g., coal combustion, biomass burning and vehicle exhaust) are important sources and should be considered seriously in future for the improvement of air quality. In my opinion, few studies simultaneously reported 15N signatures for both $NH_4^+$ and $NO_3^-$ and I think this one-year dataset is valuable and probably will improve our knowledge on the sources of air pollution. I support its publication after some minor revisions.

AC1: Thanks for your recognition of our work and for providing professional comments and valuable suggestions. These comments and suggestions are valuable and helpful for improving our manuscript. We have made revisions based on these comments (The detailed corrections are marked in the revised manuscript). If you have any further comments and suggestions, we will try our best to improve our manuscript.

RC2: Line 66-68: The dominant source of atmospheric $NH_3$ highly depends on the scale of study area. For example, the dominant emitter of $NH_3$ in the whole China should be the agricultural source; while the dominant emitter may be the vehicular emission for a city site. Therefore, cautions need to be taken when you describe this sentence.

AC2: Thanks for your professional comments. We agree with you that the dominant emitter of $NH_3$ in the whole China should be the agricultural source; while the dominant emitter may be the vehicular emission for a city site. In addition, there is a potential impact of biomass burning in suburban areas on urban $NH_3$. In general, biomass burning activity increases during autumn in Central China. Xiao et al. found that biomass burning contributed $34.5 \pm 20.4\%$, $46.4 \pm 21.4\%$, and $40.4 \pm 17.4\%$ to $NH_4^+$ for three urban sites Nanchang, Wuhan, and Changsha, respectively, during autumn(Xiao et al., 2020). The combustion sources in Lines 66-68 represent coal combustion, vehicle emission, and biomass burning. Now, we have rewritten this sentence, as shown in the

marked revised manuscript **lines 75-78**: Biomass burning in the suburbs also has a potential impact on urban $NH_3$(Xiao et al., 2020). As for urban $NH_3$, combustion sources (including coal combustion, vehicles emission, and biomass burning) were gradually becoming dominant sources in recent years verified by $\delta^{15}N$-NHx $(NH_3+NH_4^+)$(Xiao et al., 2020; Pan et al., 2018).

RC3: Line 122-126: Many $\delta^{15}N$-$NH_3$ endmembers of sources were collected by passive samplers. Did you correct these values when you conducted the source apportionment? Also, the endmembers and source numbers are important parameters for $\delta^{15}N$-derived source apportionment model and I suggest you add these in the main manuscript.

AC3: Thanks for your kind suggestion. We considered and corrected the difference of $\delta^{15}N$-$NH_3$ values resulting by passive samplers. The $\delta^{15}N$-$NH_3$ values collected by passive samplers were significantly lower than that of the active sampler, with a difference of $15.4 \pm 3.5$‰(Pan et al., 2020). The $\delta^{15}N$ of $NH_3$ from fertilizer, livestock, and urban waste collected by passive sampler(Chang et al., 2016; Felix et al., 2013; Bhattarai et al., 2020) were corrected using $15.4 \pm 3.5$‰ (Bhattarai et al., 2021; Pan et al., 2020). In addition, we have added the parameters of $\delta^{15}N$ of $NH_3$ from different sources, as shown in **line 212** (**Table 1** in marked manuscript).

**Referee#2_ Table 1 (Table 1** in manuscript). The estimation of $\delta^{15}N$-$NH_3$ and $\delta^{15}N$-NOx from various sources.

| Source | $\delta^{15}N$-$NH_3$(‰) | References |
|---|---|---|
| Biomass burning | 17.5±7.8 | (Kawashima and Kurahashi, 2011; Xiao et al., 2020) |
| Coal combustion | -2.5±6.4 | (Felix et al., 2013; Pan et al., 2016) |
| Urban traffic | 6.6±2.1 | (Walters et al., 2020) |
| Fertilizer | -28.3±5.8 | (Bhattarai et al., 2021; Chang et al., 2016; Felix et al., 2013; Bhattarai et al., 2020) |
| Livestock | -18.3±7.7 | (Bhattarai et al., 2021; Chang et al., 2016; Felix et al., 2013; Bhattarai et al., 2020) |
| Urban waste | -22.8±3.6 | (Bhattarai et al., 2021; Chang et al., 2016) |
| Source | $\delta^{15}N$-NOx(‰) | References |
| Biomass burning | 1.04±4.13 | (Zong et al., 2017; Fibiger and Hastings, 2016; Zong et al., 2022) |
| Coal combustion | 13.72±4.57 | (Zong et al., 2017; Felix et al., 2015; Felix et al., |

| | | 2012) |
|---|---|---|
| Mobile source | -7.25±7.80 | (Zong et al., 2017; Walters et al., 2015) |
| Soil microbial process | -33.77±12.16 | (Zong et al., 2017; Felix and Elliott, 2013) |

RC4: Line 149: Fig 1. Can you please highlight/mark the seasonal periods in this figure? I think this will improve the readability because you mentioned the seasonal values.

AC4: Thanks for your kind suggestion. We have marked the season in **Figure 1**, as shown in the marked manuscript **line 236**.

[Figure]

**Referee#2_Figure 1(Figure 1 in manuscript).** The concentration and $\delta^{15}N$ of $NH_4^+$ (a) and concentration, $\delta^{15}N$, and $\delta^{18}O$ of $NO_3^-$ (b).

RC5: Line 157: "average+"?

AC5: We apologize for the confusion caused by "average+". The plus symbol ("+") means positive number. Now we have deleted the + symbol, as shown in the marked manuscript **lines 245-246**.

RC6: Line 163: It would be better to provide the way you got the NH₃ concentration in the main manuscript.

AC6: We are sorry for that we don't measure the $NH_3$ concentration. The proportion of the initial $NH_3$ converted to $NH_4^+$ (f, $NH_4^+/(NH_3+NH_4^+)$) for different months referenced from a previous study in Guangzhou(Liao et al., 2014).

RC7: Line 238/274 (Fig2, Fig3): again, please highlight/mark the seasonal periods (spring, summer, autumn, and winter).

AC7: Thanks for your kind suggestion. We have marked the season in Figure 2 and Figure 3, as shown in the marked manuscript **line 329** and **line 368**.

[Figure]

**Referee#2_Figure 2 (Figure 2 in manuscript).** The sources apportionment results of atmospheric $NH_4^+$ (a) and $NO_3^-$ (b) in Guangzhou, and the comparison of sources results between $NH_4^+$ and $NO_3^-$ (c).

[Figure]

**Referee#2_Figure 3 (Figure 3 in manuscript)**. The contribution of the OH radical oxidation and $N_2O_5$ hydrolysis pathway to $NO_3^-$ (a). The vertical position of the dots corresponded to the contribution of $N_2O_5$ pathway and the size of the dots corresponded to the concentration of $NO_3^-$ (b).

RC8: Line 291-292: Why you defined BeiChengHuang Island and Heshan as the sites receiving strong anthropogenic impact? These two sites are not located in cities and should be impacted less anthropogenic activities than megacities such as Beijing and Guangzhou.

AC8: Firstly, we apologize for the wrong place name "BeiChengHuang Island". It should be BeiHuangCheng Island. We have revised it in the marked manuscript. Secondly, we doubtless agree with you that these two sites are not located in cities. However, in winter, 74% of the air mass in Beihuangcheng Island come directly from the heavily polluted Beijing-Tianjin-Hebei region(Zong, 2017). And about 26% of the air mass reached Beihuangcheng Island from the Beijing-Tianjin-Hebei region through the Shandong Peninsula(Zong, 2017). Zong et al. reported that coal combustion, mobile source, and biomass burning contributed 86.3% to $NO_3^-$ in Beihuangcheng Island, as

shown in the following figure (Zong et al., 2017). The Heshan Atmospheric Environment Monitoring Superstation is a rural station located 50 km southwest of Guangzhou (Xu et al., 2022). During the winter northeast-monsoon season, Heshan site well intercepts high anthropogenically dominated outflow airmass from Chinese continental(Xu et al., 2022). The anthropogenic sources (including fossil and biomass burning) accounted for 78% of total oxalic acid, tracers of aqueous secondary organic aerosol, in the continental outflow samples(Xu et al., 2022). Su et al. reported that coal combustion, mobile source, and biomass burning contributed 90.6% to $NO_3^-$ in Heshan (Su et al., 2020). Therefore, $NO_3^-$ was predominantly derived from anthropogenic sources in Beihuangcheng island and Heshan.

[Figure]

**Referee#2_Figure 4**. Contributions of coal combustion, mobile source, biomass burning, and biogenic soil emissions for NOx in different seasons on Beihuangcheng Island. (Zong et al., 2017)

RC9: Line 311-313. This explanation sounds reasonable. I suggest you add the references to support the facts you mentioned here (stricter vehicle emission standard, promotion of new electric vehicles etc.).

AC9: We appreciate your explicit suggestion. In order to continuously improve the Guangdong province's ambient air quality, the Guangdong Provincial Government formulated the Guangdong Air Pollution Prevention and Control Action Plan (2014-2017). The plan includes in-depth promotion of power plant pollution reduction, comprehensive promotion of boiler pollution remediation, raising the environmental

standard of new vehicles, acceleration the improvement of gasoline and diesel quality, etc(Guangdongprovince, 2014). Especially in Guangzhou and Shenzhen, clean energy vehicles will account for more than 60% of annual new buses from 2014 (Guangdongprovince, 2014). In addition, China introduced an ultra-low emissions (ULE) standards policy for renovating coal-fired power-generating units in 2014. Tang et al., found that between 2014 and 2017 China's annual power emissions of NOx dropped by 60% since the implementation of ULE policy (Tang et al., 2019). Now, we have added the above references to the marked manuscript **line 408**.

RC10: Line 324-325: "The contribution of biomass burning and vehicle was stable through a year." The vehicular emission, in my opinion, is likely constant because people drive cars in all seasons. However, the biomass burning activity generally is highly related with seasons. Can you make some explanations on this?

AC10: Thanks for your insightful comment and kind suggestion. We totally agree with you that biomass burning is highly related to the seasons. Generally, high intensity biomass burning occurred in winter in Guangdong province (dry season, i.e., from November to March)(Xu et al., 2019). $K^+$ is a typical tracer of biomass burning. The concentration of $K^+$ enhanced in winter ($0.4\mu g/m^3$) was higher than that in summer($0.2\mu g/m^3$) and autumn($0.2\mu g/m^3$), respectively, indicating enhancement of biomass burning intensity. Also, $NO_3^-$ concentration of biomass burning remarkably enhanced in winter ($1.2\mu g/m^3$) and was higher than that in summer ($0.4\mu g/m^3$) and autumn ($0.3\mu g/m^3$), respectively. However, coal combustion also enhanced in winter due to the demand for heating in North China. Our sampling site was influenced by the air mass with high coal combustion contribution from the North by long-range transportation, which may reduce the contribution of biomass burning relatively. Thus, the contribution of biomass burning showed stable compared with coal combustion. We have added the explanation in the marked manuscript, **lines 421-431**.

References:

Bhattarai, N., Wang, S., Pan, Y., Xu, Q., Zhang, Y., Chang, Y., and Fang, Y.: $\delta^{15}$N-stable isotope analysis of NHx : An overview on analytical measurements, source sampling and its source apportionment, Front. Environ. Sci. Eng., 15, 126, https://doi.org/10.1007/s11783-021-1414-6, 2021.

Bhattarai, N., Wang, S., Xu, Q., Dong, Z., Chang, X., Jiang, Y., and Zheng, H.: Sources of gaseous $NH_3$ in urban Beijing from parallel sampling of $NH_3$ and $NH_4^+$, their nitrogen isotope measurement and modeling, Sci. Total Environ., 747, 141361, https://doi.org/10.1016/j.scitotenv.2020.141361, 2020.

Chang, Y., Liu, X., Deng, C., Dore, A. J., and Zhuang, G.: Source apportionment of atmospheric ammonia before, during, and after the 2014 APEC summit in Beijing using stable nitrogen isotope signatures, Atmos. Chem. Phys., 16, 11635-11647, https://doi.org/10.5194/acp-16-11635-2016, 2016.

Felix, J. D. and Elliott, E. M.: The agricultural history of human-nitrogen interactions as recorded in ice core $\delta^{15}$N-$NO_3^-$, Geophys. Res. Lett., 40, 1642-1646, https://doi.org/10.1002/grl.50209, 2013.

Felix, J. D., Elliott, E. M., and Shaw, S. L.: Nitrogen isotopic composition of coal-fired power plant NOx: influence of emission controls and implications for global emission inventories, Environ. Sci. Technol., 46, 3528-3535, https://doi.org/10.1021/es203355v, 2012.

Felix, J. D., Elliott, E. M., Gish, T. J., McConnell, L. L., and Shaw, S. L.: Characterizing the isotopic composition of atmospheric ammonia emission sources using passive samplers and a combined oxidation-bacterial denitrifier approach, Rapid Commun. Mass Spectrom., 27, 2239-2246, https://doi.org/10.1002/rcm.6679, 2013.

Felix, J. D., Elliott, E. M., Avery, G. B., Kieber, R. J., Mead, R. N., Willey, J. D., and Mullaugh, K. M.: Isotopic composition of nitrate in sequential Hurricane Irene precipitation samples: Implications for changing NOx sources, Atmos. Environ., 106, 191-195, https://doi.org/10.1016/j.atmosenv.2015.01.075, 2015.

Fibiger, D. L. and Hastings, M. G.: First Measurements of the Nitrogen Isotopic Composition of NOx from Biomass Burning, Environ. Sci. Technol., 50, 11569-11574, https://doi.org/10.1021/acs.est.6b03510, 2016.

Action Plan for Air Pollution Control of Guangdong Province (2014-2017): http://www.gd.gov.cn/gkmlpt/content/0/142/mpost_142687.html, last access: February 14, 2014.

Kawashima, H. and Kurahashi, T.: Inorganic ion and nitrogen isotopic compositions of atmospheric aerosols at Yurihonjo, Japan: implications for nitrogen sources, Atmos. Environ., 45, 6309-6316, https://doi.org/10.1016/j.atmosenv.2011.08.057, 2011.

Liao, B., Wu, D., Chang, Y., Lin, Y., Wang, S., and Li, F.: Characteristics of particulate $SO_4^{2-}$, $NO_3^-$, $NH_4^+$, and related gaseous pollutants in Guangzhou (in Chinese), Acta Sci. Circumst., 34, 1551-1559, https://doi.org/10.13671/j.hjkxxb.2014.0218, 2014.

Pan, Y., Tian, S., Liu, D., Fang, Y., Zhu, X., Zhang, Q., Zheng, B., Michalski, G., and Wang, Y.: Fossil fuel combustion-related emissions dominate atmospheric ammonia sources during severe haze episodes: evidence from $^{15}$N-stable isotope in size-resolved aerosol ammonium, Environ. Sci. Technol. 50, 8049-8056, https://doi.org/10.1021/acs.est.6b00634, 2016.

Pan, Y., Tian, S., Liu, D., Fang, Y., Zhu, X., Gao, M., Wentworth, G. R., Michalski, G., Huang, X., and Wang, Y.: Source Apportionment of Aerosol Ammonium in an Ammonia-Rich Atmosphere: An Isotopic Study of Summer Clean and Hazy Days in Urban Beijing, J. Geophys. Res.: Atmos., 123, 5681-5689, https://doi.org/10.1029/2017jd028095, 2018.

Pan, Y., Gu, M., Song, L., Tian, S., Wu, D., Walters, W. W., Yu, X., Lü, X., Ni, X., Wang, Y., Cao, J., Liu, X., Fang, Y., and Wang, Y.: Systematic low bias of passive samplers in characterizing nitrogen

isotopic composition of atmospheric ammonia, Atmos. Res., 243, https://doi.org/10.1016/j.atmosres.2020.105018, 2020.

Su, T., Li, J., Tian, C., Zong, Z., Chen, D., and Zhang, G.: Source and formation of fine particulate nitrate in South China: Constrained by isotopic modeling and online trace gas analysis, Atmos. Environ., 231, https://doi.org/10.1016/j.atmosenv.2020.117563, 2020.

Tang, L., Qu, J., Mi, Z., Bo, X., Chang, X., Anadon, L. D., Wang, S., Xue, X., Li, S., Wang, X., and Zhao, X.: Substantial emission reductions from Chinese power plants after the introduction of ultra-low emissions standards, Nat. Energy, 4, 929-938, https://doi.org/10.1038/s41560-019-0468-1, 2019.

Walters, W. W., Tharp, B. D., Fang, H., Kozak, B. J., and Michalski, G.: Nitrogen Isotope Composition of Thermally Produced NOx from Various Fossil-Fuel Combustion Sources, Environ. Sci. Technol., 49, 11363-11371, https://doi.org/10.1021/acs.est.5b02769, 2015.

Walters, W. W., Song, L., Chai, J., Fang, Y., Colombi, N., and Hastings, M. G.: Characterizing the spatiotemporal nitrogen stable isotopic composition of ammonia in vehicle plumes, Atmos. Chem. Phys., 20, 11551-11567, https://doi.org/10.5194/acp-20-11551-2020, 2020.

Xiao, H. W., Wu, J. F., Luo, L., Liu, C., Xie, Y. J., and Xiao, H. Y.: Enhanced biomass burning as a source of aerosol ammonium over cities in central China in autumn, Environ. Pollut., 266, 115278, https://doi.org/10.1016/j.envpol.2020.115278, 2020.

Xu, B., Zhang, G., Gustafsson, O., Kawamura, K., Li, J., Andersson, A., Bikkina, S., Kunwar, B., Pokhrel, A., Zhong, G., Zhao, S., Li, J., Huang, C., Cheng, Z., Zhu, S., Peng, P., and Sheng, G.: Large contribution of fossil-derived components to aqueous secondary organic aerosols in China, Nat. Commun., 13, 5115, https://doi.org/10.1038/s41467-022-32863-3, 2022.

Xu, Y., Huang, Z., Jia, G., Fan, M., Cheng, L., Chen, L., Shao, M., and Zheng, J.: Regional discrepancies in spatiotemporal variations and driving forces of open crop residue burning emissions in China, Sci. Total Environ., 671, 536-547, https://doi.org/10.1016/j.scitotenv.2019.03.199, 2019.

Zong, Z.: Composition and source apportionment of $PM_{2.5}$ at the background area in North China, Doctor, Yantai Institute of Coastal Zone Research, Chinese Academy of Sciences, 2017.

Zong, Z., Shi, X., Sun, Z., Tian, C., Li, J., Fang, Y., Gao, H., and Zhang, G.: Nitrogen isotopic composition of NOx from residential biomass burning and coal combustion in North China, Environ. Pollut., 304, 119238, https://doi.org/10.1016/j.envpol.2022.119238, 2022.

Zong, Z., Wang, X., Tian, C., Chen, Y., Fang, Y., Zhang, F., Li, C., Sun, J., Li, J., and Zhang, G.: First assessment of NOx sources at a regional background site in North China using isotopic analysis linked with modeling, Environ. Sci. Technol., 51, 5923-5931, https://doi.org/10.1021/acs.est.6b06316, 2017.

---

## Author Response (AR1)

**Title: High contribution of anthropogenic combustion sources to atmospheric inorganic reactive nitrogen in South China evidenced by isotopes**

**Manuscript ID: acp-2023-49**

Dear editor:

Thank you for your letter.

We appreciate the two referees for their professional comments on our manuscript. We considered these detailed comments and responded to their suggestions and questions. Based on referees' comments, we have carefully revised our manuscript. The following is one-on-one response to the reviewers for your reference. Revised texts are marked in red in our manuscript.

We sincerely appreciate your consideration. If there are any questions, please contact us. We will do our best to solve it.

With Best Regards,

Jun Li junli@gig.ac.cn

**Response to Referee #1**

RC- Reviewer's Comments; AC – Authors' Response Comments

RC1: The manuscript by Li et al. simultaneously reported concentrations and stable nitrogen isotope and oxygen isotopes compositions of atmospheric $NO_3^-$ and concentrations and nitrogen isotopes compositions of atmospheric $NH_4^+$ in $PM_{2.5}$ samples collected in Guangzhou from May 2017 to June 2018. Then, authors restrained nitrogen isotope fractionation values of the process of $NH_3$ to formed $NH_4^+$ and $NO_x$ to formed $NO_3^-$. Finally, using the IsoSource model, authors quantified the relative contributions of major sources of $NH_3$ and $NO_x$ to atmospheric $NH_4^+$ and $NO_3^-$, respectively. Authors found the focus of $NH_3$ reduction should be on anthropogenic combustion sources especially on biomass burning, which might be responsible for the lag of the decline in deposition of air pollutions behind the reduction in emission. Additionally, despite a series of measures to reduce emissions of $NO_x$, fossil fuels, as the main energy for production and living, will still inevitably emit a large amount of $NO_x$. Authors emphasized that the emission of atmospheric inorganic nitrogen is largely related to anthropogenic combustion sources. The development and promotion of clean energy and efficient use of biomass are conducive to the deep reduction of atmospheric nitrogen. I believe that this result is meaningful and would make a substantial contribution to the field. The manuscript is generally well-organized in structure. If the following comments are adequately addressed, I believe the manuscript could be accepted to Atmospheric Chemistry and Physics.

AC1: We appreciate your constructive comments and professional suggestions. These comments and suggestions are helpful for improving our manuscript. Based on your comments and suggestions, we have revised our manuscript. If you have any further comments and suggestions, we will do our best to improve our manuscript.

We would like to show the details as follows:

RC2: Lines 112-113: The author needs to provide the analytical accuracy of isotopes nitrogen and oxygen isotopes.

AC2: Thanks for your suggestion. We have added details on the accuracy of nitrogen and oxygen isotope analysis, as shown in the marked revised manuscript **lines 120-127:** To ensure the stability of the instrument, standard samples were tested for every ten samples. The standard deviation of replicates was generally less than 0.4‰, 0.8‰, and 0.5‰ for $\delta^{15}N$-$NO_3^-$, $\delta^{18}O$-$NO_3^-$, and $\delta^{15}N$-$NH_4^+$, respectively. The instrumental values of $\delta^{15}N$-$NO_3^-$ and $\delta^{18}O$-$NO_3^-$ were corrected by multi-point correction ($\delta^{18}O$ $r^2=0.99$, $\delta^{15}N$ $r^2=0.999$) based on international standards (IAEA-NO-3, USGS32, USGS34, and USGS35). The measured values of $\delta^{15}N$-$NH_4^+$ were also corrected by multi-point correction ($r^2=0.999$) based on international standards (IAEA-N1, USGS25, and USGS26).

RC3: Nitrogen isotope fractionation values of the process of $NH_3$ to formed $NH_4^+$ and NOx to formed $NO_3^-$ are key parameters for quantifying the relative contributions of major sources of $NH_3$ and NOx to atmospheric $NH_4^+$ and $NO_3^-$. The calculation methods for the two parameters should be include in the text of manuscript. In addition, it is necessary to give readers detailed data of each parameter, especially the fractionation value.

AC3: Thanks for your professional comment and kind suggestion.

**a. Nitrogen isotope fractionation values of the process of $NH_3$ to form $NH_4^+$.**

Atmospheric initial $\delta^{15}N$-$NH_3$ was calculated by following equation 1.

$$\delta^{15}N\text{-}NH_{3\text{-initial}} = \delta^{15}N\text{-}NH_4^+ - \varepsilon(NH_4^+\text{-}NH_3) \times (1-f) \qquad (1)$$

Where, $\delta^{15}N$-$NH_4^+$ and $\delta^{15}N$-$NH_{3\text{-initial}}$ represent the $\delta^{15}N$ of particulate $NH_4^+$ and atmospheric initial $NH_3$, respectively. $\varepsilon(NH_4^+$-$NH_3)$ represents the isotope fractionation factor in the gaseous $NH_3$ conversion to particulate $NH_4^+$ in the atmosphere. The f value represents the proportion of the initial $NH_3$ converted to $NH_4^+$, referring to $NH_3$ and $NH_4^+$ observed in Guangzhou (Liao et al., 2014).

The $\varepsilon(NH_4^+$-$NH_3)$ value is temperature dependent(Huang et al., 2019), which can be deduced from(Urey, 1947), as shown in equation 2. The atmospheric average temperature was 24.5°C in our sampling period, and the corresponding $\varepsilon(NH_4^+\text{-}NH_3)$ value was 34.2‰ calculated by equation 2. In addition, the $\varepsilon(NH_4^+\text{-}NH_3)$ in Guangzhou was estimated to be 32.4‰ according to equation 6. Equation 6 was deduced by equations 3-5. According to equation 6, a linear fitting equation was observed between $fNH_4^+$ and $\delta^{15}N\text{-}NH_4^+$ (**Referee#1_Figure 1**), and the absolute value of the slope (32.4‰) was equal to $\varepsilon(NH_4^+\text{-}NH_3)$. The $\varepsilon(NH_4^+\text{-}NH_3)$ average of the two methods (34.2‰ and 32.4‰) was 33.3‰ and approximated to the experimental isotope enrichment factor (33‰)(Heaton et al., 1997). Therefore, +33‰ was used for deducing the $\delta^{15}N$ of the initial $NH_3$. We have added the calculation process to manuscript. Please see **lines 137-162** in the marked revised manuscript.

$$\varepsilon_{(NH_4^+\_NH_3)} = 12.4678 * \frac{1000}{T+273.15} - 7.6694 \qquad (2)$$

$$\delta^{15}N\text{-}NH_4^+ - \delta^{15}N\text{-}NH_3 = \varepsilon_{(NH_4^+\_NH_3)} \qquad (3)$$

$$fNH_4^+ + fNH_3 = 1 \qquad (4)$$

$$\delta^{15}N\text{-}NH_4^+ * fNH_4^+ + \left(\delta^{15}N\text{-}NH_4^+ - \varepsilon_{(NH_4^+\_NH_3)}\right) * (1 - fNH_4^+) = \delta^{15}N \qquad (5)$$

$$\delta^{15}N\text{-}NH_4^+ = -\varepsilon_{(NH_4^+\_NH_3)} * fNH_4^+ + \left(\delta^{15}N + \varepsilon_{(NH_4^+\_NH_3)}\right) \qquad (6)$$

Where, T represents the atmospheric temperature (°C). $\delta^{15}N\text{-}NH_4^+$ and $\delta^{15}N\text{-}NH_3$ represent the $\delta^{15}N$ of particulate $NH_4^+$ and atmospheric $NH_3$, respectively. $\delta^{15}N$ represents the sum of $\delta^{15}N\text{-}NH_4^+$ and $\delta^{15}N\text{-}NH_3$. $fNH_3$ and $fNH_4^+$ represent the proportion of atmospheric $NH_3$ and particulate $NH_4^+$, respectively.

[Figure]

**Referee#1_Figure 1 (Figure S1 in SI).** Linear fitting of $NH_4^+/(NH_3+NH_4^+)$ with $\delta^{15}N$-

$NH_4^+$.

**b. Nitrogen isotope fractionation values of the process of $NO_x$ to form $NO_3^-$**

In Central Pearl River Delta, $NO_3^-$ formed through ·OH and $N_2O_5$ pathways contributed to 94% simulated by CAMQ model (Qu et al., 2021). In this study, only ·OH (R4) and $N_2O_5$ (R5-R7) formation pathways were considered. The reasons why we only consider the ·OH oxidation and $N_2O_5$ hydrolysis pathway to form $NO_3^-$ were explained in detail in the **AC7**.

$$NO + O_3 \rightarrow NO_2 + O_2 \qquad (R1)$$

$$NO_2 + hv \rightarrow NO + O \qquad (R2)$$

$$O + O_2 \rightarrow O_3 \qquad (R3)$$

$$NO_2 + \cdot OH \rightarrow HNO_3 \qquad (R4)$$

$$NO_2 + O_3 \rightarrow NO_3 + O_2 \qquad (R5)$$

$$NO_2 + NO_3 \rightarrow N_2O_5 \qquad (R6)$$

$$N_2O_5 + H_2O \rightarrow 2HNO_3 \qquad (R7)$$

$$HNO_3 + Alkali \rightarrow NO_3^- \qquad (R8)$$

The specific details of the Bayesian mixing model were reported by our previous studies (Zong et al., 2017; Zong et al., 2020). The principle and process of Bayesian mixing model was shown in **Referee#1_Figure 2** adapted from Zong et al., (Zong et al., 2017). The atmospheric $\delta^{18}O$-$NO_3^-$ can be expressed by equation 7. The [$\delta^{18}O$-$HNO_3$]$_{OH}$ can be further expressed by equation 8 assuming no kinetic isotope fractionation (Walters and Michalski, 2016). And [$\delta^{18}O$-$HNO_3$]$_{H2O}$ can be estimated by equation 9 (Walters and Michalski, 2016). The $\delta^{18}O$ values in tropospheric $H_2O$, $NOx$, $O_3$, and OH were within a certain range. The tropospheric $\delta^{18}O$-$H_2O$, $\delta^{18}O$-$NOx$, $\delta^{18}O$-$O_3$, and $\delta^{18}O$-OH ranged from -25‰ to 0‰(Baskaran et al., 2011; Walters and Michalski, 2016), 112‰ to 122‰ (Michalski et al., 2014; Walters and Michalski, 2016), 90‰ to 122‰, and -15‰ to 0‰, respectively(Fang et al., 2011; Johnston and Thiemens, 1997). Therefore, the γ (the contribution of ·OH formation pathway) can be estimated by $f$$NO_2$ and oxygen isotope fractionation i.e., α$NO_2$/NO, αOH/$H_2O$, and α$N_2O_5$/$NO_2$. The oxygen isotope fractionations are temperature dependent and can be estimated by equation 11. The $f$$NO_2$ varied from 0.20 to 0.95(Zong et al., 2017; Walters et al., 2016).

Based on $\delta^{18}O$-NO$_3^-$, $\delta^{18}O$-H$_2$O, $\delta^{18}O$-NOx, $\delta^{18}O$-O$_3$, and temperature (equations 7-11, **Referee#1_Table 1**), $\gamma$ (maximum $\gamma$ and minimum $\gamma$) was estimated by Monte Carlo simulation nested in Bayesian mixing model (Zong et al., 2017). Assuming no kinetic isotope fractionation, the nitrogen isotope fractionation value in the formation process of NO$_3^-$ ($\epsilon$N) was calculated by equations 11-14 combined with $\gamma$ and temperature (Zong et al., 2017; Walters and Michalski, 2016; Walters et al., 2016). The $\epsilon$N value in our sampling period was 5.1±2.5‰, which was comparable to that in Beijing(average 6.5‰)(Fan et al., 2020). The contributions of different sources to atmospheric NOx were quantified by Bayesian mixing model coupled with $\epsilon$N, $\delta^{15}N$-atmospheric-NO$_3^-$, and $\delta^{15}N$-NOx endmembers. We have added the methods in the marked revised manuscript, **lines 169-211.**

$$\delta^{18}O\text{-}NO_3^- = \gamma \times [\delta^{18}O\text{-}NO_3^-]_{OH} + (1-\gamma) \times [\delta^{18}O\text{-}NO_3^-]_{H_2O} = \gamma \times [\delta^{18}O\text{-}HNO_3]_{OH} + (1-\gamma) \times [\delta^{18}O\text{-}HNO_3]_{H_2O} \tag{7}$$

$$[\delta^{18}O\text{-}HNO_3]_{OH} = \frac{2}{3}[(\delta^{18}O\text{-}NO_2)]_{OH} + \frac{1}{3}[\delta^{18}O\text{-}OH]_{OH} = \frac{2}{3}\left[\frac{1000\times(^{18}\alpha_{NO_2/NO}-1)(1-f_{NO_2})}{(1-f_{NO_2})+(^{18}\alpha_{NO_2/NO}\times f_{NO_2})} + [\delta^{18}O\text{-}NO_X]\right] + \frac{1}{3}[(\delta^{18}O\text{-}H_2O) + 1000\times(^{18}\alpha_{OH/H_2O}-1)] \tag{8}$$

$$[\delta^{18}O\text{-}HNO_3]_{H_2O} = \frac{5}{6}(\delta^{18}O\text{-}N_2O_5) + \frac{1}{6}(\delta^{18}O\text{-}H_2O) \tag{9}$$

$$\delta^{18}O\text{-}N_2O_5 = \delta^{18}O\text{-}NO_2 + 1000\times(^{18}\alpha_{N_2O_5/NO_2}-1) \tag{10}$$

$$1000(^m\alpha_{X/Y}-1) = \frac{A}{T^4}\times 10^{10} + \frac{B}{T^3}\times 10^8 + \frac{C}{T^2}\times 10^6 + \frac{D}{T}\times 10^4 \tag{11}$$

$$\epsilon N = \gamma \times \epsilon(\delta^{15}N\text{-}NO_3^-)_{OH} + (1-\gamma) \times \epsilon(\delta^{15}N\text{-}NO_3^-)_{H_2O}$$
$$= \gamma \times \epsilon(\delta^{15}N\text{-}HNO_3)_{OH} + (1-\gamma) \times \epsilon(\delta^{15}N\text{-}HNO_3)_{H_2O} \tag{12}$$

$$\epsilon(\delta^{15}N\text{-}HNO_3)_{OH} = \epsilon(\delta^{15}N\text{-}NO_2)_{OH} = 1000\times\left[\frac{(^{15}\alpha_{NO_2/NO}-1)(1-f_{NO_2})}{(1-f_{NO_2})+(^{15}\alpha_{NO_2/NO}\times f_{NO_2})}\right] \tag{13}$$

$$\epsilon(\delta^{15}N\text{-}HNO_3)_{H_2O} = \epsilon(\delta^{15}N\text{-}N_2O_5)_{H_2O} = 1000\times(^{15}\alpha_{N_2O_5/NO_2}-1) \tag{14}$$

Where, γ is the contribution of ·OH formation pathway to $NO_3^-$, εN is the nitrogen isotope fractionation value. $fNO_2$ is the fraction of $NO_2$ in the total NOx. $^{18}\alpha NO_2/NO$, $^{18}\alpha OH/H_2O$, $^{18}\alpha N_2O_5/NO_2$ are the oxygen isotope equilibrium fractionation factors between $NO_2$ and NO, ·OH and $H_2O$, $N_2O_5$ and $NO_2$, respectively. $^{15}\alpha NO_2/NO$ and $^{15}\alpha N_2O_5/NO_2$ are the nitrogen isotope equilibrium fractionation factor between $NO_2$ and NO, $N_2O_5$ and $NO_2$, respectively.

[Figure]

**Referee#1_Figure 2.** Principle and process of Bayesian mixing model in this study, the "E" represents equation in the following section, "εN" refers to N fractionation, and "SIR" is "sampling-importance-resampling"(Zong et al., 2017).

**Referee#1_Table 1 (Table S1 in SI).** Test constants of A, B, C, and D over the settled temperature range of 150−450K(Zong et al., 2017; Walters and Michalski, 2016; Walters and Michalski, 2015; Walters et al., 2016).

| $^m\alpha_{X/Y}$ | A | B | C | D |
|---|---|---|---|---|
| $^{15}NO_2/NO$ | 3.8834 | -7.7299 | 6.0101 | -0.17928 |
| $^{15}N_2O_5/NO_2$ | 0.69398 | -1.9859 | 2.3876 | 0.16308 |
| $^{18}NO/NO_2$ | -0.04129 | 1.1605 | -1.8829 | 0.74723 |
| $^{18}H_2O/OH$ | 2.1137 | -3.8026 | 2.5653 | 0.59410 |

RC4: Authors should explain why these four sources are selected as main sources of atmospheric $NO_3^-$ and these six sources are selected as main sources of atmospheric $NH_4^+$?

AC4:Thanks for your comment. The following was the explanation for our selection of sources of atmospheric $NO_3^-$ and $NH_4^+$. We have also added the explanations in **SI Text S2.**

  **a.** We considered coal combustion, mobile traffic sources, biomass burning, and soil microbial activity as dominant atmospheric NOx sources. Based on bottom-up emission inventory, power plant, industry, residential use, and transportation were the traditional NOx emission sources in cities in China, including Guangzhou (Liu et al., 2017). According to the type of fuel combustion, traditional sources of NOx could be roughly divided into coal combustion (power plant, industry, and residential use) and mobile sources (transportation including vehicle exhaust and ship emission). Furthermore, recent studies show that biomass burning is an essential source of NOx based on emission factor study (Mehmood et al., 2017) and isotopic evidence (Zong et al., 2020). Microbial process emission is another important source of NOx, in which nitrification or denitrification microbial bacteria widely distributed in soils consume accumulated nitrogen and release NO as a byproduct(Hall and Matson, 1996; Jaeglé et al., 2004). The cultivated land with extensive use of nitrogen fertilizer in the suburbs around Guangzhou is also an important source of NOx, which is named as microbial process in this study. $\delta^{15}N$-NOx values differed significantly among these four sources, which allows us to differentiate their relative contributions to the mixture of atmospheric. We did not consider $NO_3^-$ from lightning because it accounts for less than 5% of global terrestrial NOx emissions(Song et al., 2021; Qu et al., 2020; Pickering et al., 2016).

  **b.** There are two major groups of atmospheric $NH_3$ emission sources(Chen et al., 2022). One is $NH_3$ volatilization from $NH_4^+$-containing substrates (mainly fertilized and natural soils, livestock, human wastes, and natural and N-polluted water). Although Guangzhou is an urban site, the emission inventory results showed a high contribution of nitrogen fertilizers application and livestock to atmospheric $NH_3$ (Zheng et al., 2012), which may be influenced by agricultural activities around Guangzhou. Human waste is also an important contributor to $NH_3$ in cities, as suggested by a study in Shanghai(Chang et al., 2015). Guangzhou is one of China's megacities with a dense population, so the contribution of human waste to atmospheric $NH_3$ in Guangzhou cannot be ignored. Therefore, nitrogen fertilizers application, livestock, and human waste were considered as sources of volatilization $NH_3$ in this study. In addition, the other group is $NH_3$ associated with combustion sources (such as coal burning, vehicles, and biomass burning). The contribution of biomass burning and coal combustion to $NH_3$ was very high (about 76.3%) in developing countries, suggested by the global high-resolution emissions inventory (Meng et al., 2017). $NH_3$ in Chinese cities was indeed influenced by coal and biomass combustion evidenced by isotopes(Xiao et al., 2020; Liu et al., 2018; Pan et al., 2018). Selective catalytic reduction technology equipped with vehicles and industrial boiler is also an important source of $NH_3$(Meng et al., 2017). With the rapid increase in vehicle ownership, vehicle emission has a significant impact on urban $NH_3$, which was confirmed by tunnel tests in Guangzhou (Liu et al., 2014). Therefore, biomass burning, coal combustion, and vehicles were considered as sources of combustion $NH_3$ in this study.

RC5: Lines 176-178: Does the combustion of sugarcane leaf emit $NH_4^+$ directly or emit $NH_3$ and then formed $NH_4^+$?

AC5: We have no field measurements of smoke and particulate matter released by sugarcane burning. Gases such as $NH_3$, NOx, and HCN can be released during biomass burning (Zhou et al., 2006; Stubenberger et al., 2008). Therefore, we speculate that $NH_3$ was released directly from the burning of sugarcane leaves, and then converted into $NH_4^+$ by atmospheric aging. Now, we have rewritten lines 176-178. The new sentence was shown in the marked manuscript **lines 267-269:** The $\delta^{15}N$ of $NH_4^+$ formed from $NH_3$ released by sugarcane leaves burning was 44.1‰ (SI Text S3), which was consistent with the highest $\delta^{15}N$-$NH_4^+$ values (45.5‰ and 45.1‰) in July.

RC6: Lines 236-237: The sources apportionment results of atmospheric $NO_3^-$ in Figure c does not correspond to that in Figure b.

AC6: We are sorry for making this mistake. Thanks for your reminding. The colors in Figure 2a and 2b do not match the previous colors in Figure 2c. Now, we have corrected this error as shown below and in the marked manuscript, **line 329**.

[Figure]

**Referee#1_Figure 3 (Figure 2 in manuscript).** The sources apportionment results of atmospheric $NH_4^+$ (a) and $NO_3^-$ (b) in Guangzhou, and the comparison of sources results between $NH_4^+$ and $NO_3^-$ (c).

RC7: Lines 272-273: Why does the author only consider the OH radical oxidation and $N_2O_5$ hydrolysis pathway to $NO_3^-$, and not consider other pathways? The author needs to explain.

AC7: Thanks for your comment and suggestion.

There are several major formation pathways of $NO_3^-$.

P1 ($NO_2+\cdot OH$), $NO_2$ is oxidized by $\cdot OH$ to form $HNO_3$, then reacts with alkaline substances (such as $NH_3$) to form $NO_3^-$.

P2 ($N_2O_5$), $NO_2$ is oxidized by $O_3$ to form $\cdot NO_3$, $\cdot NO_3$ reacts with $NO_2$ to form $N_2O_5$, then the hydrolysis of $N_2O_5$ on aerosol surfaces produces $NO_3^-$.

P3 ($\cdot NO_3+org$), the $NO_2$ is oxidized by $O_3$ to form $\cdot NO_3$, then the $\cdot NO_3$ reacts with organic, such as dimethyl sulfide (DMS) or hydrocarbons (HC) to form $HNO_3$, and then $NO_3^-$.

P4($\cdot NO_3+\cdot HO_2$), $NO_2$ is oxidized by $O_3$ to form $\cdot NO_3$, $\cdot NO_3$ reacts with $\cdot HO_2$ to form $HNO_3$.

The P1 ($\cdot OH$) and P2 ($N_2O_5$) pathways are dominant formation pathways. Song reported that $\cdot OH$ and $N_2O_5$ pathways contributed 43% and 32% to $NO_3^-$, respectively, by isotope tracing (Song et al., 2021). Based on isotopic estimates, the contribution of $\cdot NO3+org$ to $NO_3^-$ was relatively high, e.g., about 16% in Beijing(Song et al., 2021). However, the proportion of $\cdot NO_3+org$ estimated by the Community Multiscale Air Quality (CAMQ) model was very low in the YRD(Sun et al., 2022) and PRD(Qu et al., 2021), especially in Guangzhou (central PRD) where it is only 4%(Qu et al., 2021). The $\cdot OH$ and $N_2O_5$ were the dominant pathways and contributed 94% to $NO_3^-$ in Guangzhou (Qu et al., 2021). We speculate that the different contribution of $\cdot NO_3+org$ pathway between Guangzhou and Beijing may be caused by the difference in atmospheric oxidation. The ozone pollution is serious in Guangzhou due to a unique synoptic system including the surface high-pressure system, hurricane movement, and sea–land breeze(Tan et al., 2019). And the atmospheric $\cdot OH$ reactivity in Guangzhou was higher than in several cities, including Beijing (Tan et al., 2019). Take DMS as an example, the main oxidant of DMS is $\cdot OH$ (Andreae and Crutzen, 1997). However, in the cold season or remote regions, the $\cdot NO_3$ radical can also play an important role in reaction with DMS (addition reaction and hydrogen abstraction) (Andreae and Crutzen, 1997; Yin et al., 1990). The high reactivity of $\cdot OH$ may reduce the contribution of $\cdot NO_3$ to DMS in Guangzhou due to the competition between $\cdot OH$ and $\cdot NO_3$ to react with DMS. Therefore, the contribution of $\cdot NO_3+org$ to $NO_3^-$ was relatively low. In addition, the $\delta^{18}O$ of $NO_3^-$ formed by the $N_2O_5$ and $\cdot NO_3+org$ pathway is similar(Walters and Michalski, 2016). The introduction of the $\cdot NO_3+org$ pathway would greatly increase the uncertainty of the contribution of $N_2O_5$ pathways. While the $\delta^{18}O$ of $NO_3^-$ formed by the $\cdot OH$ and $N_2O_5$ pathway differ significantly, which allows to differentiate their relative contributions to $NO_3^-$. Therefore, we only considered the $\cdot OH$ and $N_2O_5$ pathways in this study. We have also added the explanation in **SI text S2**.


**Response to Referee #2**

RC- Reviewer's Comments; AC – Authors' Response Comments

RC1: This paper estimated the relative contributions of main sources to ammonium and nitrate aerosols in a subtropical megacity of South China using stable N isotope analysis. They found that anthropogenic activities (e.g., coal combustion, biomass burning and vehicle exhaust) are important sources and should be considered seriously in future for the improvement of air quality. In my opinion, few studies simultaneously reported 15N signatures for both $NH_4^+$ and $NO_3^-$ and I think this one-year dataset is valuable and probably will improve our knowledge on the sources of air pollution. I support its publication after some minor revisions.

AC1: Thanks for your recognition of our work and for providing professional comments and valuable suggestions. These comments and suggestions are valuable and helpful for improving our manuscript. We have made revisions based on these comments (The detailed corrections are marked in the revised manuscript). If you have any further comments and suggestions, we will try our best to improve our manuscript.

RC2: Line 66-68: The dominant source of atmospheric $NH_3$ highly depends on the scale of study area. For example, the dominant emitter of $NH_3$ in the whole China should be the agricultural source; while the dominant emitter may be the vehicular emission for a city site. Therefore, cautions need to be taken when you describe this sentence.

AC2: Thanks for your professional comments. We agree with you that the dominant emitter of $NH_3$ in the whole China should be the agricultural source; while the dominant emitter may be the vehicular emission for a city site. In addition, there is a potential impact of biomass burning in suburban areas on urban $NH_3$. In general, biomass burning activity increases during autumn in Central China. Xiao et al. found that biomass burning contributed $34.5 \pm 20.4\%$, $46.4 \pm 21.4\%$, and $40.4 \pm 17.4\%$ to $NH_4^+$ for three urban sites Nanchang, Wuhan, and Changsha, respectively, during autumn(Xiao et al., 2020). The combustion sources in Lines 66-68 represent coal combustion, vehicle emission, and biomass burning. Now, we have rewritten this sentence, as shown in the marked revised manuscript **lines 75-78**: Biomass burning in the suburbs also has a potential impact on urban $NH_3$(Xiao et al., 2020). As for urban $NH_3$, combustion sources (including coal combustion, vehicles emission, and biomass burning) were gradually becoming dominant sources in recent years verified by $\delta^{15}N$-NHx $(NH_3+NH_4^+)$(Xiao et al., 2020; Pan et al., 2018).

RC3: Line 122-126: Many $\delta^{15}N$-$NH_3$ endmembers of sources were collected by passive samplers. Did you correct these values when you conducted the source apportionment? Also, the endmembers and source numbers are important parameters for $\delta^{15}N$-derived source apportionment model and I suggest you add these in the main manuscript.

AC3: Thanks for your kind suggestion. We considered and corrected the difference of $\delta^{15}N$-$NH_3$ values resulting by passive samplers. The $\delta^{15}N$-$NH_3$ values collected by passive samplers were significantly lower than that of the active sampler, with a difference of $15.4 \pm 3.5$‰(Pan et al., 2020). The $\delta^{15}N$ of $NH_3$ from fertilizer, livestock, and urban waste collected by passive sampler(Chang et al., 2016; Felix et al., 2013; Bhattarai et al., 2020) were corrected using $15.4 \pm 3.5$‰ (Bhattarai et al., 2021; Pan et al., 2020). In addition, we have added the parameters of $\delta^{15}N$ of $NH_3$ from different sources, as shown in **line 212** (**Table 1** in marked manuscript).

**Referee#2_ Table 1 (Table 1** in manuscript). The estimation of $\delta^{15}N$-$NH_3$ and $\delta^{15}N$-NOx from various sources.

| Source | $\delta^{15}N$-$NH_3$(‰) | References |
|---|---|---|
| Biomass burning | 17.5±7.8 | (Kawashima and Kurahashi, 2011; Xiao et al., 2020) |
| Coal combustion | -2.5±6.4 | (Felix et al., 2013; Pan et al., 2016) |
| Urban traffic | 6.6±2.1 | (Walters et al., 2020) |
| Fertilizer | -28.3±5.8 | (Bhattarai et al., 2021; Chang et al., 2016; Felix et al., 2013; Bhattarai et al., 2020) |
| Livestock | -18.3±7.7 | (Bhattarai et al., 2021; Chang et al., 2016; Felix et al., 2013; Bhattarai et al., 2020) |
| Urban waste | -22.8±3.6 | (Bhattarai et al., 2021; Chang et al., 2016) |
| Source | $\delta^{15}N$-NOx(‰) | References |
| Biomass burning | 1.04±4.13 | (Zong et al., 2017; Fibiger and Hastings, 2016; Zong et al., 2022) |
| Coal combustion | 13.72±4.57 | (Zong et al., 2017; Felix et al., 2015; Felix et al., |

| | | 2012) |
|---|---|---|
| Mobile source | -7.25±7.80 | (Zong et al., 2017; Walters et al., 2015) |
| Soil microbial process | -33.77±12.16 | (Zong et al., 2017; Felix and Elliott, 2013) |

RC4: Line 149: Fig 1. Can you please highlight/mark the seasonal periods in this figure? I think this will improve the readability because you mentioned the seasonal values.

AC4: Thanks for your kind suggestion. We have marked the season in **Figure 1**, as shown in the marked manuscript **line 236**.

[Figure]

**Referee#2_Figure 1(Figure 1 in manuscript).** The concentration and $\delta^{15}N$ of $NH_4^+$ (a) and concentration, $\delta^{15}N$, and $\delta^{18}O$ of $NO_3^-$ (b).

RC5: Line 157: "average+"?

AC5: We apologize for the confusion caused by "average+". The plus symbol ("+") means positive number. Now we have deleted the + symbol, as shown in the marked manuscript **lines 245-246**.

RC6: Line 163: It would be better to provide the way you got the NH₃ concentration in the main manuscript.

AC6: We are sorry for that we don't measure the NH₃ concentration. The proportion of the initial NH₃ converted to $NH_4^+$ (f, $NH_4^+/(NH_3+NH_4^+)$) for different months referenced from a previous study in Guangzhou(Liao et al., 2014).

RC7: Line 238/274 (Fig2, Fig3): again, please highlight/mark the seasonal periods (spring, summer, autumn, and winter).

AC7: Thanks for your kind suggestion. We have marked the season in Figure 2 and Figure 3, as shown in the marked manuscript **line 329** and **line 368**.

[Figure]

**Referee#2_Figure 2 (Figure 2 in manuscript).** The sources apportionment results of atmospheric $NH_4^+$ (a) and $NO_3^-$ (b) in Guangzhou, and the comparison of sources results between $NH_4^+$ and $NO_3^-$ (c).

[Figure]

**Referee#2_Figure 3 (Figure 3 in manuscript)**. The contribution of the OH radical oxidation and $N_2O_5$ hydrolysis pathway to $NO_3^-$ (a). The vertical position of the dots corresponded to the contribution of $N_2O_5$ pathway and the size of the dots corresponded to the concentration of $NO_3^-$ (b).

RC8: Line 291-292: Why you defined BeiChengHuang Island and Heshan as the sites receiving strong anthropogenic impact? These two sites are not located in cities and should be impacted less anthropogenic activities than megacities such as Beijing and Guangzhou.

AC8: Firstly, we apologize for the wrong place name "BeiChengHuang Island". It should be BeiHuangCheng Island. We have revised it in the marked manuscript. Secondly, we doubtless agree with you that these two sites are not located in cities. However, in winter, 74% of the air mass in Beihuangcheng Island come directly from the heavily polluted Beijing-Tianjin-Hebei region(Zong, 2017). And about 26% of the air mass reached Beihuangcheng Island from the Beijing-Tianjin-Hebei region through the Shandong Peninsula(Zong, 2017). Zong et al. reported that coal combustion, mobile source, and biomass burning contributed 86.3% to $NO_3^-$ in Beihuangcheng Island, as shown in the following figure (Zong et al., 2017). The Heshan Atmospheric Environment Monitoring Superstation is a rural station located 50 km southwest of Guangzhou (Xu et al., 2022). During the winter northeast-monsoon season, Heshan site well intercepts high anthropogenically dominated outflow airmass from Chinese continental(Xu et al., 2022). The anthropogenic sources (including fossil and biomass burning) accounted for 78% of total oxalic acid, tracers of aqueous secondary organic aerosol, in the continental outflow samples(Xu et al., 2022). Su et al. reported that coal combustion, mobile source, and biomass burning contributed 90.6% to $NO_3^-$ in Heshan (Su et al., 2020). Therefore, $NO_3^-$ was predominantly derived from anthropogenic sources in Beihuangcheng island and Heshan.

[Figure]

**Referee#2_Figure 4**. Contributions of coal combustion, mobile source, biomass burning, and biogenic soil emissions for NOx in different seasons on Beihuangcheng Island. (Zong et al., 2017)

RC9: Line 311-313. This explanation sounds reasonable. I suggest you add the references to support the facts you mentioned here (stricter vehicle emission standard, promotion of new electric vehicles etc.).

AC9: We appreciate your explicit suggestion. In order to continuously improve the Guangdong province's ambient air quality, the Guangdong Provincial Government formulated the Guangdong Air Pollution Prevention and Control Action Plan (2014-2017). The plan includes in-depth promotion of power plant pollution reduction, comprehensive promotion of boiler pollution remediation, raising the environmental standard of new vehicles, acceleration the improvement of gasoline and diesel quality, etc(Guangdongprovince, 2014). Especially in Guangzhou and Shenzhen, clean energy vehicles will account for more than 60% of annual new buses from 2014 (Guangdongprovince, 2014). In addition, China introduced an ultra-low emissions (ULE) standards policy for renovating coal-fired power-generating units in 2014. Tang et al., found that between 2014 and 2017 China's annual power emissions of NOx dropped by 60% since the implementation of ULE policy (Tang et al., 2019). Now, we have added the above references to the marked manuscript **line 408**.

RC10: Line 324-325: "The contribution of biomass burning and vehicle was stable through a year." The vehicular emission, in my opinion, is likely constant because people drive cars in all seasons. However, the biomass burning activity generally is highly related with seasons. Can you make some explanations on this?

AC10: Thanks for your insightful comment and kind suggestion. We totally agree with you that biomass burning is highly related to the seasons. Generally, high intensity biomass burning occurred in winter in Guangdong province (dry season, i.e., from November to March)(Xu et al., 2019). $K^+$ is a typical tracer of biomass burning. The concentration of $K^+$ enhanced in winter ($0.4\mu g/m^3$) was higher than that in summer($0.2\mu g/m^3$) and autumn($0.2\mu g/m^3$), respectively, indicating enhancement of biomass burning intensity. Also, $NO_3^-$ concentration of biomass burning remarkably enhanced in winter ($1.2\mu g/m^3$) and was higher than that in summer ($0.4\mu g/m^3$) and autumn ($0.3\mu g/m^3$), respectively. However, coal combustion also enhanced in winter due to the demand for heating in North China. Our sampling site was influenced by the air mass with high coal combustion contribution from the North by long-range transportation, which may reduce the contribution of biomass burning relatively. Thus, the contribution of biomass burning showed stable compared with coal combustion. We have added the explanation in the marked manuscript, **lines 421-431**.


$$\varepsilon_{(NH_4^+\_NH_3)} = 12.4678 * \frac{1000}{T+273.15} - 7.6694 \qquad (4)$$

$$\delta^{15}N\text{-}NH_4^+ - \delta^{15}N\text{-}NH_3 = \varepsilon_{(NH_4^+\_NH_3)} \qquad (5)$$

$$fNH_4^+ + fNH_3 = 1 \qquad (6)$$

$$\delta^{15}N\text{-}NH_4^+ * fNH_4^+ + \left(\delta^{15}N\text{-}NH_4^+ - \varepsilon_{(NH_4^+\_NH_3)}\right) * (1 - fNH_4^+) = \delta^{15}N \quad (7)$$

$$\delta^{15}N\text{-}NH_4^+ = -\varepsilon_{(NH_4^+\_NH_3)} * fNH_4^+ + (\delta^{15}N + \varepsilon_{(NH_4^+\_NH_3)}) \qquad (8)$$

Where, T represents the atmospheric temperature (°C). $\delta^{15}N\text{-}NH_4^+$ and $\delta^{15}N\text{-}NH_3$ represent the $\delta^{15}N$ of particulate $NH_4^+$ and atmospheric $NH_3$, respectively. $\delta^{15}N$ represents the sum of $\delta^{15}N\text{-}NH_4^+$ and $\delta^{15}N\text{-}NH_3$. $fNH_3$ and $fNH_4^+$ represent the proportion of atmospheric $NH_3$ and particulate $NH_4^+$, respectively.

**Bayesian mixing model.** $\delta^{15}N$ were used for tracing source based on conservation of isotopic mass. Bayesian mixing model improved upon linear mixing models by explicitly considering uncertainty in prior information and isotopic equilibrium fractionation. Recently, Bayesian mixing model was applied to trace the sources of atmospheric pollutants(Zong et al., 2017; Zong et al., 2020). The model coupled with $\delta^{15}N\text{-}NO_3^-$ and $\delta^{18}O\text{-}NO_3^-$ were used to identify the formation process and quantify the sources contribution of $NO_3^-$.

In Central Pearl River Delta (PRD), $NO_3^-$ formed through ·OH and $N_2O_5$ pathways contributed to 94% simulated by CAMQ model (Qu et al., 2021). In this study, only ·OH and $N_2O_5$ formation pathways were considered. Details of $NO_3^-$ formation pathway were also shown in **SI Text S2**. The atmospheric $\delta^{18}O\text{-}NO_3^-$ can be expressed by Eq. (9). The [δ18O-

HNO₃]_OH can be further expressed by Eq. (10) assuming no kinetic isotope fractionation (Walters and Michalski, 2016). And $[\delta^{18}O\text{-}HNO_3]_{H_2O}$ can be estimated by Eq. (11) (Walters and Michalski, 2016). The $\delta^{18}O$ values in tropospheric $H_2O$, NOx, $O_3$, and OH were within a certain range. The tropospheric $\delta^{18}O\text{-}H_2O$, $\delta^{18}O\text{-}NOx$, $\delta^{18}O\text{-}O_3$, and $\delta^{18}O\text{-}OH$ ranged from -25‰ to 0‰(Baskaran et al., 2011; Walters and Michalski, 2016), 112‰ to 122‰ (Michalski et al., 2014; Walters and Michalski, 2016), 90‰ to 122‰, and -15‰ to 0‰, respectively(Fang et al., 2011; Johnston and Thiemens, 1997). Therefore, the γ (the contribution of ·OH formation pathway) can be estimated by $f$NO₂ and oxygen isotope fractionation i.e., αNO₂/NO, αOH/H₂O, and αN₂O₅/NO₂. The oxygen isotope fractionations are temperature dependent and can be estimated by Eq. (13) and **Table S1.** The $f$NO₂ varied from 0.20 to 0.95(Zong et al., 2017; Walters et al., 2016). Based on $\delta^{18}O\text{-}NO_3^-$, $\delta^{18}O\text{-}H_2O$, $\delta^{18}O\text{-}NOx$, $\delta^{18}O\text{-}O_3$, and temperature (Eq. (9-13)), γ (maximum γ and minimum γ) was estimated by Monte Carlo simulation nested in Bayesian mixing model (Zong et al., 2017). Assuming no kinetic isotope fractionation, the nitrogen isotope fractionation value in the formation process of $NO_3^-$ (εN) was calculated by Eq. (13-16) combined with γ and temperature (Zong et al., 2017; Walters and Michalski, 2016; Walters et al., 2016). The εN value in our sampling period was 5.1±2.5‰, which was comparable to that in Beijing(average 6.5‰)(Fan et al., 2020). The contributions of different sources to atmospheric NOx were quantified by Bayesian mixing model coupled with εN, $\delta^{15}N$-atmospheric-$NO_3^-$, and $\delta^{15}N\text{-}NOx$ endmembers shown in **Table 1**. We considered coal combustion, mobile traffic sources, biomass burning, and soil microbial process as dominant atmospheric NOx sources in Guangzhou, details shown in **SI Text S2**. The specific details of Bayesian mixing model were reported by our previous studies(Zong et al., 2017; Zong et al., 2020).

$$\delta^{18}O\text{-}NO_3^- = \gamma \times [\delta^{18}O\text{-}NO_3^-]_{OH} + (1-\gamma) \times [\delta^{18}O\text{-}NO_3^-]_{H_2O} = \gamma \times [\delta^{18}O\text{-}HNO_3]_{OH} +$$

$$(1-\gamma) \times [\delta^{18}O\text{-}HNO_3]_{H_2O} \hspace{4cm} (9)$$

$$[\delta^{18}O\text{-}HNO_3]_{OH} = \frac{2}{3}[(\delta^{18}O\text{-}NO_2)]_{OH} + \frac{1}{3}[\delta^{18}O\text{-}OH]_{OH} = \frac{2}{3}\left[\frac{1000 \times (^{18}\alpha_{NO_2/NO}-1)(1-f_{NO_2})}{(1-f_{NO_2})+(^{18}\alpha_{NO_2/NO}\times f_{NO_2})} + \right.$$

$$\left.[\delta^{18}O\text{-}NO_X]\right] + \frac{1}{3}[(\delta^{18}O\text{-}H_2O) + 1000 \times (^{18}\alpha_{OH/H_2O}-1)] \hspace{1.5cm} (10)$$

$$[\delta^{18}O\text{-}HNO_3]_{H_2O} = \frac{5}{6}(\delta^{18}O\text{-}N_2O_5) + \frac{1}{6}(\delta^{18}O\text{-}H_2O) \hspace{2cm} (11)$$

$$\delta^{18}O\text{-}N_2O_5 = \delta^{18}O\text{-}NO_2 + 1000 \times \left({}^{18}\alpha_{N_2O_5/NO_2} - 1\right) \hspace{2cm} (12)$$

$$1000({}^{m}\alpha_{X/Y} - 1) = \frac{A}{T^4} \times 10^{10} + \frac{B}{T^3} \times 10^8 + \frac{C}{T^2} \times 10^6 + \frac{D}{T} \times 10^4 \hspace{1cm} (13)$$

$$\varepsilon N = \gamma \times \varepsilon(\delta^{15}N\text{-}NO_3^-)_{OH} + (1 - \gamma) \times \varepsilon(\delta^{15}N\text{-}NO_3^-)_{H_2O}$$
$$= \gamma \times \varepsilon(\delta^{15}N\text{-}HNO_3)_{OH} + (1 - \gamma) \times \varepsilon(\delta^{15}N\text{-}HNO_3)_{H_2O} \hspace{1cm} (14)$$

$$\varepsilon(\delta^{15}N\text{-}HNO_3)_{OH} = \varepsilon(\delta^{15}N\text{-}NO_2)_{OH} = 1000 \times \left[\frac{({}^{15}\alpha_{NO_2/NO}-1)(1-f_{NO_2})}{(1-f_{NO_2})+({}^{15}\alpha_{NO_2/NO} \times f_{NO_2})}\right] (15)$$

$$\varepsilon(\delta^{15}N\text{-}HNO_3)_{H_2O} = \varepsilon(\delta^{15}N\text{-}N_2O_5)_{H_2O} = 1000 \times \left({}^{15}\alpha_{N_2O_5/NO_2} - 1\right) \hspace{0.5cm} (16)$$

[revised manuscript text omitted]
 Sources of atmospheric $NH_3$ and NOx in Guangzhou, $NO_3^-$ formation pathways in Guangzhou**

**Atmospheric $NH_3$ sources.** There are two major groups of atmospheric $NH_3$ emission sources(Chen et al., 2022b). One is $NH_3$ volatilization from $NH_4^+$-containing substrates (mainly fertilized and natural soils, livestock, human wastes, and natural and N-polluted water). Although Guangzhou is an urban site, the emission inventory results showed a high contribution of nitrogen fertilizers application and livestock to atmospheric $NH_3$ (Zheng et al., 2012), which may be influenced by agricultural activities around Guangzhou. Human waste is also an important contributor to $NH_3$ in cities, as suggested by a study in Shanghai(Chang et al., 2015). Guangzhou is one of China's megacities with a dense population, so the contribution of human waste to atmospheric $NH_3$ in Guangzhou cannot be ignored. Therefore, nitrogen fertilizers application, livestock, and human waste were considered as sources of volatilization $NH_3$ in this study. In addition, the other group is $NH_3$ associated with combustion sources (such as coal burning, vehicles, and biomass burning). The contribution of biomass burning and coal combustion to $NH_3$ was very high (about 76.3%) in developing countries, as suggested by the global high-resolution emissions inventory (Meng et al., 2017). $NH_3$ in Chinese cities was indeed influenced by coal and biomass combustion evidenced by isotopes(Xiao et al., 2020; Liu et al., 2018; Pan et al., 2018). Selective catalytic reduction technology equipped with vehicles and industrial boiler is also an important source of $NH_3$(Meng et al., 2017). With the rapid increase in vehicle ownership, vehicle emission has a significant impact on urban $NH_3$, which was confirmed by tunnel tests in Guangzhou (Liu et al., 2014). Therefore, biomass burning, coal combustion, and vehicles were considered as sources of combustion $NH_3$ in this study.

**Atmospheric NOx sources.** We considered coal combustion, mobile traffic sources, biomass burning, and soil microbial activity as dominant atmospheric NOx sources. Based on bottom-up emission inventory, power plant, industry, residential use, and transportation were the traditional NOx emission sources in cities in China, including Guangzhou (Liu et al., 2017a). According to the type of fuel combustion, traditional sources of NOx could be roughly divided into coal combustion (power plant, industry, and residential use) and mobile sources (transportation including vehicle exhaust and ship emission). Furthermore, recent studies show that biomass burning is an essential source of NOx based on emission factor study (Mehmood et al., 2017) and isotopic evidence (Zong et al., 2020). Microbial process emission is another important source of NOx, in which nitrification or denitrification microbial bacteria widely distributed in soils consume accumulated nitrogen and release NO as a byproduct(Hall and Matson, 1996; Jaeglé et al., 2004). The cultivated land with extensive use of nitrogen fertilizer in the suburbs around Guangzhou is also an important source of NOx, which is named as microbial process in this study. $\delta^{15}N$-NOx values differed significantly among these four sources, which allows us to differentiate their relative contributions to the mixture of atmospheric. We did not consider $NO_3^-$ from lightning because it accounts for less than 5% of global terrestrial NOx emissions(Song et al.,

2021; Qu et al., 2020; Pickering et al., 2016).

**$NO_3^-$ formation pathways.** There are several major formation pathways of $NO_3^-$.

P1 ($NO_2$+·OH), $NO_2$ is oxidized by ·OH to form $HNO_3$, then reacts with alkaline substances (such as $NH_3$) to form $NO_3^-$.

P2 ($N_2O_5$), $NO_2$ is oxidized by $O_3$ to form ·$NO_3$, ·$NO_3$ reacts with $NO_2$ to form

$N_2O_5$, then the hydrolysis of $N_2O_5$ on aerosol surfaces produces $NO_3^-$.

P3 (·$NO_3$+org), the $NO_2$ is oxidized by $O_3$ to form ·$NO_3$, then the ·$NO_3$ reacts with organic, such as dimethyl sulfide (DMS) or hydrocarbons (HC) to form $HNO_3$, and then

$NO_3^-$.

P4(·$NO_3$+·$HO_2$), $NO_2$ is oxidized by $O_3$ to form ·$NO_3$, ·$NO_3$ reacts with ·$HO_2$ to form $HNO_3$.

The P1 ($\cdot OH$) and P2 ($N_2O_5$) pathways are dominant formation pathways. Song reported that $\cdot OH$ and $N_2O_5$ pathways contributed 43% and 32% to $NO_3^-$, respectively, by isotope tracing (Song et al., 2021). Based on isotopic estimates, the contribution of $\cdot NO3+org$ to $NO_3^-$ was relatively high, e.g., about 16% in Beijing(Song et al., 2021).

However, the proportion of $\cdot NO_3+org$ estimated by the Community Multiscale Air

Quality (CAMQ) model was very low in the YRD(Sun et al., 2022) and PRD(Qu et al.,

2021), especially in Guangzhou (central PRD) where it is only 4%(Qu et al., 2021).

The $\cdot OH$ and $N_2O_5$ were the dominant pathways and contributed 94% to $NO_3^-$ in

Guangzhou (Qu et al., 2021). We speculate that the different contribution of $\cdot NO_3+org$

pathway between Guangzhou and Beijing may be caused by the difference in atmospheric oxidation. The ozone pollution is serious in Guangzhou due to a unique synoptic system including the surface high-pressure system, hurricane movement, and sea–land breeze(Tan et al., 2019). And the atmospheric $\cdot OH$ reactivity in Guangzhou was higher than in several cities, including Beijing (Tan et al., 2019). Take DMS as an example, the main oxidant of DMS is $\cdot OH$ (Andreae and Crutzen, 1997). However, in the cold season or remote regions, the $\cdot NO_3$ radical can also play an important role in reaction with DMS (addition reaction and hydrogen abstraction) (Andreae and Crutzen,

1997; Yin et al., 1990). The high reactivity of $\cdot OH$ may reduce the contribution of $\cdot NO_3$

to DMS in Guangzhou due to the competition between $\cdot OH$ and $\cdot NO_3$ to react with

DMS. Therefore, the contribution of $\cdot NO_3+org$ to $NO_3^-$ was relatively low. In addition, the $\delta^{18}O$ of $NO_3^-$ formed by the $N_2O_5$ and $\cdot NO_3+org$ pathway is similar(Walters and

Michalski, 2016). The introduction of the $\cdot NO_3+org$ pathway would greatly increase the uncertainty of the contribution of $N_2O_5$ pathways. While the $\delta^{18}O$ of $NO_3^-$ formed by the $\cdot OH$ and $N_2O_5$ pathway differ significantly, which allows to differentiate their relative contributions to $NO_3^-$. Therefore, we considered only the $\cdot OH$ and $N_2O_5$

pathways in this study.

Specifically, the $\cdot OH$ and $N_2O_5$ pathways are expressed by R1-R8. Once emitted into the atmosphere, NOx is oxidized to $HNO_3$ or $NO_3^-$ via the following chemical pathways (R1-R8) (Fang et al., 2011). In summary, NOx oxygen atoms are rapidly exchanged with $O_3$ in the NO/$NO_2$ cycle (R1-R3); ·OH radicals result in the oxidation of $NO_2$ to $HNO_3$ (R4; the ·OH pathway); $NO_2$ is oxidized by $O_3$ to produce ·$NO_3$ (R5), which subsequently combines with $NO_2$ to form $N_2O_5$ (R6), and then undergoes hydrolysis to form $HNO_3$ (R7), referred to as the $O_3$ pathway; and the generated $HNO_3$ combines with alkali to form $NO_3^-$ (R8). Overall, the ·OH and $O_3$ pathways are the two fundamental oxidation pathways for NOx, generally exhibiting noticeable diurnal and seasonal variation(Elliott et al., 2007). Previous research has found that the ·OH pathway is more prevalent during the daytime and in summer when the relative concentration of ·OH is higher. Conversely, the $O_3$ pathway is more dominant overnight and in winter, because $N_2O_5$ is thermally unstable(Hastings et al., 2003; Xiao et al., 2015). The $O_3$ in the troposphere has a higher $\delta^{18}O$ value, while $\delta^{18}O$-OH and $\delta^{18}O$-$H_2O$ is lower. The $\delta^{18}O$-$HNO_3$ formed by the ·OH pathway is contributed by 2/3 $O_3$ and 1/3 ·OH (R4), while in the $N_2O_5$ hydrolysis pathway after oxidation by $O_3$, the $\delta^{18}O$-$HNO_3$ is contributed by 5/6 $O_3$ and 1/6 $H_2O$ (R5-R7). Therefore, the $\delta^{18}O$-$NO_3^-$ formed through the ·OH pathway is lower than the $N_2O_5$ pathway.

$NO + O_3 \rightarrow NO_2 + O_2$ ____(R1)

$NO_2 + hv \rightarrow NO + O$ ____(R2)

$O + O_2 \rightarrow O_3$ __________(R3)

$NO_2 + \cdot OH \rightarrow HNO_3$ ______(R4)

$NO_2 + O_3 \rightarrow NO_3 + O_2$ ___(R5)

$NO_2 + NO_3 \rightarrow N_2O_5$ ______(R6)

$N_2O_5 + H_2O \rightarrow 2HNO_3$ ___(R7)

$HNO_3 + Alkali \rightarrow NO_3^-$ ___(R8)

**Text S3 The estimation of $\delta^{15}N$-$NH_4^+$ from sugarcane leaf burning**

The $\delta^{15}N$ in sugarcane leaf is 38‰ (Martinellia et al., 2002), which may consist of N-$NO_X$ and N-$NH_3$. The $\delta^{15}N$-NOx from biomass burning is 1.04‰(Zong et al., 2017). According to the assumption of different proportions (from 5% to 95%) of N-$NO_X$ and N-$NH_3$ from sugarcane leaf, shown in Table S32. The mean value among the proportion (from 5% to 95%) of N-$NH_4^+$ in sugarcane leaf was 37.48‰. In addition, the $\delta^{15}N$ of particulate matters from biomass burning was 6.6‰ higher than that of biomass (Martinellia et al., 2002). Therefore, $\delta^{15}N\text{-}NH_4^+$ from sugarcane leaf burning may be

44.08‰.

**Figure S1-Figure S6**

[Figure]

**Figure S2S1.** Linear fitting of $NH_4^+/(NH_3+NH_4^+)$ with $\delta^{15}N-NH_4^+$.

[Figure]

**Figure S12.** Ranges of $\delta^{15}N-NH_4^+$ from different sites(Pan et al., 2016; Kundu et al.,

2010; Kawashima and Kurahashi, 2011) and different emission sources(Felix et al.,

2013; Bhattarai et al., 2021; Chang et al., 2016; Xiao et al., 2020).

[Figure]

**Figure S3.** The comparison of sources apportionment results of atmospheric NH₃ and NH₄⁺ in different sites in China. Background site in Tai mountain[TMt] (Wu et al., 2021; Chang et al., 2019), urban sites in North China (Beijing [BJ] (Pan et al., 2020; Chang et al., 2016), vertical profile observation in Beijing (ground, 120m height, and 260m height [BJ-ground, BJ-120m, and BJ-260m] (Wu et al., 2019), Jingjinji region [BTH] (Zhang et al., 2020), and North China plain [NCP]) (Xiang et al., 2022), East North China (Harbin heating period and non-heating period [HRB-H and HRB-NH]) (Sun et al., 2021), Central China (Wuhan [WH] and Changsha [CS]) (Xiao et al., 2020), East China (Nangchang [NC]) (Xiao et al., 2020), Southwest China (Guiyang [GY], source in precipitation) (Liu et al., 2017b), and South China (Guangzhou[GZ]) (Liu et al., 2018), vertical profile observation in Guangzhou( ground and Guangzhou tower [GZ-ground and GZ-tower])(Chen et al., 2022a). Source of NH₃ were estimated by inventory methods in developing country[developing] (Meng et al., 2017).

[Figure]

**Figure S4.** The air mass backward trajectory to receptor site on 7 July,2017 and 25

Jan,2018.

[Figure]

**Figure S5.** The temporal variation of $O_3$ concentration in PRD from 2013 to 2021.

[Figure]

Figure S6. The comparison of sources apportionment results of atmospheric NOx and
NO₃⁻ in different sites in China. Background in  Beihuangcheng island [ BH
island](Zong et al., 2017) and Tai mountain [TMt] (Wu et al., 2021), urban sites in
North China (Beijing [BJ] (Zong et al., 2020), Beijing winter [BJ-W] (Fan et al., 2020),
and vertical profile observation in Beijing [BJ-tower](Fan et al., 2022)), Central China
(Wuhan [WH]) (Zong et al., 2020), East China (Shanghai [SH]) (Zong et al., 2020),
Southwest China (Chengdu [CD]), and South China (Guangzhou [GZ2014] and Heshan
[HS])(Zong et al., 2020; Su et al., 2020).

**Table S1-Table S2**

**Table S1**. Test constants of A, B, C, and D over the settled temperature range of 150−450K(Zong et al., 2017; Walters and Michalski, 2016; Walters et al., 2016; Walters and Michalski, 2015).

| $^m\alpha_{X/Y}$ | A | B | C | D |
|---|---|---|---|---|
| $^{15}NO_2/NO$ | 3.8834 | -7.7299 | 6.0101 | -0.17928 |
| $^{15}N_2O_5/NO_2$ | 0.69398 | -1.9859 | 2.3876 | 0.16308 |
| $^{18}NO/NO_2$ | -0.04129 | 1.1605 | -1.8829 | 0.74723 |
| $^{18}H_2O/OH$ | 2.1137 | -3.8026 | 2.5653 | 0.59410 |

**Table S32.** The estimation of $\delta^{15}N\text{-}NH_3$ in sugarcane leaf.

| N-NOx in sugarcane leaf (%) | 5 | 25 | 50 | 75 | 95 |
|---|---|---|---|---|---|
| $\delta^{15}N$ in sugarcane leaf (‰) | 38 | 38 | 38 | 38 | 38 |
| $\delta^{15}N\text{-}NOx$ (‰) | 1.04 | 1.04 | 1.04 | 1.04 | 1.04 |
| Caculated results $\delta^{15}N\text{-}NH_3$ (‰) | 37.95 | 37.74 | 37.48 | 37.22 | 37.01 |